# DOES INSTRUCTION TUNING REDUCE DIVERSITY?
# A CASE STUDY USING CODE GENERATION

## ABSTRACT

Large Language Models (LLMs) should ideally generate diverse content for *open-ended prompts*. Preliminary evidence has suggested that preference-tuned language models struggle to generate diverse content, which would have important implications for how we align models. However, research on this question has been limited by the difficulty of measuring diversity, which naïvely would require costly human evaluation. We propose to leverage code as a means to study semantic diversity since code has executable semantics. To this end, we create an open-ended program synthesis task, enabling us to cheaply evaluate the diversity of hundreds of thousands of generations. Using our methodology, we find that while preference-tuning reduces syntactic and lexical diversity, it can increase semantic diversity. We also study the effect of model size and prompting technique on diversity. Finally, we find that neural diversity metrics correlate poorly with our semantic diversity metrics, highlighting the need for more rigorous methodologies for evaluating diversity.

## 1 INTRODUCTION

As large language models (LLMs) become powerful at a wide variety of tasks (Zhao et al., 2024; Zheng et al., 2024), it is important to evaluate the diversity of their generations, not just their accuracy. Many real-world tasks are open-ended to some degree with many possible answers—e.g., crafting convincing essays, suggesting cooking recipes, writing unit tests, etc. Evaluating diversity can also provide insights into the nature of language models, especially their creative capabilities. Additionally, diversity is of paramount importance to the *exploration* component of algorithms such as Reinforcement Learning from Human Feedback (**RLHF**).

One of the key research questions explores how different kinds of instruction tuning impact diversity. In particular, recent work has demonstrated that RLHF may reduce diversity in summarization with small ALPACAFARM models (Kirk et al., 2023) and in joke generation with CHATGPT (Jentzsch & Kersting, 2023). Furthermore, it has been hypothesized that RLHF induces "mode collapse" on a broader scale (Kirk et al., 2023; janus, 2022). These results may impact decisions about which algorithms to use for instruction tuning.

However, defining diversity is a key challenge in addressing this and related questions. Linguistics distinguishes between *content* (or *semantics*)—the meaning of an utterance—and *form*, which refers to how that meaning is conveyed. Diversity of form can be measured automatically using lexical metrics, such as $n$-gram diversity (Li et al., 2016).

Unfortunately, measuring the diversity of content/semantics is challenging since it typically requires human evaluation, which can be prohibitively expensive to scale and highly subjective from one individual to another. One strategy is using neural models as proxies for human evaluation; however, recent work has shown that neural methods are inferior to human evaluation (Tevet & Berant, 2021).

In this work, we propose to address this challenge by evaluating diversity in the task of open-ended program generation. Since programs have well-defined, executable semantics, we can automatically evaluate semantic diversity reliably. In particular, we can define three separate forms of diversity for a given program (and a fixed set of test cases): its *lexical form* is reflected by the sequence of tokens in a program, its *syntactic form* is reflected by the Abstract Syntax Tree, and its *semantics* is reflected by the outputs for given test case inputs. Correspondingly, we can measure lexical diversity using

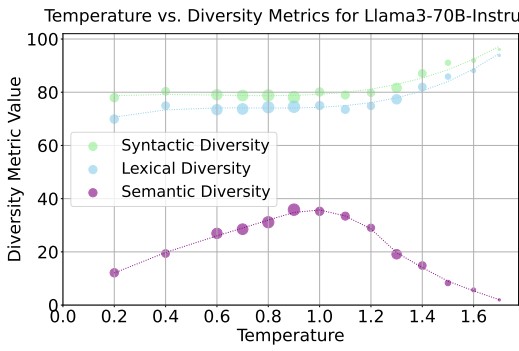

Figure 1: Left: Diversity metrics when modulating the Temperature parameter for LLAMA3-INSTRUCT 70B. Metrics measuring lexical or syntactic diversity increase with temperature, even as high-temperature outputs become incoherent. Right: Two Python programs generated by CODEL-LAMA 7B with different implementations, demonstrating high syntactic diversity and significant lexical diversity despite being semantically equivalent in our dataset.

.

metrics such as $n$-gram bag of words, syntactic diversity using extensions of $n$-grams to trees, and semantic diversity by comparing executed program outputs. Crucially, evaluating semantic diversity is objective and involves executing generated programs, enabling us to scale diversity evaluation to hundreds of thousands of generations from various configurations and prompts. In Figure 1 we demonstrate how our execution-based semantic diversity captures the phenomenon of a "sweet-spot" for sampling with temperature: as temperature increases, generations become more diverse. Until eventually at high temperatures they become degenerate and incoherent. This behavior is not captured by standard lexical and syntactic diversity metrics.

To implement this strategy, we create a dataset of open-ended program synthesis tasks and evaluate the impact of different instruction tuning techniques on different kinds of diversity using this dataset. In particular, we consider models that are instruction-tuned using both *supervised fine-tuning* (**SFT**) and *preference-tuning* techniques (i.e., combinations of PPO, DPO, and Rejection Sampling). We find that, in general, compared to base models, instruction tuning reduces the lexical and syntactic diversity of generations while increasing their semantic diversity; this effect is especially pronounced in preference-tuned models but persists for SFT models. In addition, we find that neural diversity metrics are a poor proxy for the actual execution semantics of generated programs, highlighting the importance of rigorously measuring semantic diversity. Finally, we analyze the impact of model size and prompting techniques on diversity.

In summary, our contributions are as follows:

- A novel methodology and dataset for evaluating diversity by focusing on programs where semantics can be disentangled from syntactic and lexical forms.
- Empirical results validating that our semantic diversity metric is not captured by neural, lexical, or syntactic diversity metrics.
- Empirical results assessing the impact of different instruction-tuning strategies on semantic, lexical, and syntactic diversity. In particular, we find that instruction tuning reduces the lexical and syntactic diversity of generations but increases the semantic diversity of generations, which paints a more nuanced picture than prior work.

## 2 BACKGROUND, RELATED WORK, AND MOTIVATION

**LLMs and Instruction Tuning Methods**. Neural language models (Bengio et al., 2000; Radford et al., 2019) are remarkably powerful, and some of the earlier efforts to align them with instruction-following relied on few-shot prompting (Brown, 2020). Later, RLHF using Proximal Policy Optimization (**PPO**) (Schulman et al., 2017; Stiennon et al., 2020) was shown to be highly effective in aligning LLMs with human preference (Ouyang et al., 2022). Subsequently, numerous alterna-

tive methods to PPO have been proposed, such as Direct Preference Optimization **(DPO)** (Rafailov et al., 2024) and Rejection Sampling (Touvron et al., 2023), which need not be mutually exclusive. Additionally, language models have shown significant promise in programming tasks and software engineering (Chen et al., 2021; Roziere et al., 2023).

**Evaluating Language Models for Code**. Evaluating the code generation abilities of neural models has roots in the semantic parsing literature (Oda et al., 2015; Yin et al., 2018), which relied on $n$-gram similarity metrics like BLEU (Papineni et al., 2002). Numerous executable programming benchmarks have been proposed, especially given that $n$-gram similarity does not imply correctness (Hendrycks et al., 2021; Chen et al., 2021; Puri et al., 2021; Li et al., 2022). However, diversity has not been a major consideration in the context of programming tasks.

**Approaches to Measuring Diversity.** Traditional methods for automatically measuring diversity fall into lexical or neural approaches. Lexical metrics generally involve calculating summary statistics using $n$-grams, such as **Distinct-N** (Li et al., 2016; Du & Black, 2019) and **Self-BLEU** (a modified BLEU metric) (Zhu et al., 2018). More recently, deep learning has been used to model the similarity of natural language sentences (Zhang et al., 2019; Reimers & Gurevych, 2019; Neelakantan et al., 2022) and code (Feng et al., 2020; Zhou et al., 2023; Guo et al., 2022; Liu et al., 2023; Zhuo, 2024). These techniques have been adapted to measuring diversity in natural language (Lai et al., 2020; Tevet & Berant, 2021; Stasaski & Hearst, 2022), but not in code. Tevet & Berant (2021) compare lexical and neural models, demonstrating that while neural metrics outperform lexical ones, they still fall short of human performance. Shaib et al. (2024) show that lexical diversity often captures the same information as traditional compression algorithms and recommend reporting a combination of lexical and neural diversity scores for a more comprehensive evaluation.

**Diversity of LLM Content**. Empirical work evaluating the diversity and creativity of LLM generations is relatively sparse. McCoy et al. (2023) investigate whether smaller language models exhibit linguistic novelty by evaluating combinations of $n$-grams not present in the training corpus. With the advent of more capable LLMs, it has been argued that models such as CHATGPT are incapable of generating diverse jokes Jentzsch & Kersting (2023). Additionally, Kirk et al. (2023) contend that RLHF induces "mode collapse", using both lexical and neural metrics to measure the diversity of summarized content. Furthermore, (Padmakumar & He, 2023) show that human-written essays assisted by an RLHF-tuned model are less diverse than those assisted by a base model when assessed using neurodiversity measures.

The dominant narrative in this body of research suggests that preference-tuned LLMs reduce diversity. This is problematic because diversity is paramount for reinforcement learning (RL) and preference tuning, as exploration is necessary to improve any RL system (Sutton & Barto, 2018). Additionally, useful assistants should be capable of producing diverse outputs in open-ended domains, ranging from brainstorming and creative writing to software testing, drafting website front-ends, mining data for insights, and scientific discovery. Diversity is also critical in addressing ambiguity, such as when responding to a question like "How can I add new functionality to my code?" These use cases are not captured by programming benchmarks that typically assume a single correct answer. The empirical question of diversity in LLM-generated content is crucial for building better RL-inspired systems and creating agents capable of handling real-world, open-ended tasks. However, current benchmarks fail to reflect this. Our best methods for automatically measuring the diversity of newer and more powerful LLMs still rely on $n$-gram-based metrics or on smaller and weaker neural models to evaluate stronger ones.

## 3 DIVERSITY EVALUATION METHODOLOGY

We describe our methodology for evaluating diversity. First, we create a dataset of problem descriptions, $D = \{x_i\}_{i=1}^n$, where each problem is designed to prompt a given LLM to generate a diverse set of programs. All programs must be executable in a standard way; we provide a detailed description of our dataset below. Next, we use the following approach to evaluate a given generative model, $f(y \mid x)$. For each problem description $x_i$ in our dataset, we generate $K$ programs, $P_i = \{p_i^1, p_i^2, ..., p_i^K\} \sim_{\text{i.i.d.}} f(\cdot \mid x_i)$. We then calculate a diversity score, $\text{Div}_m(P_i)$, where $m$ denotes the diversity metric used (see Section 4 for the metrics we employ). Finally, we compute

---

**Input Description:**
- Multiple datasets.
- Each dataset consists of four real numbers: a, b, c, d.
- There are no more than 30 datasets.

**Example Input:**
```
35.68 139.77 51.15 359.82
01.37 103.92 41.78 272.25
51.15 359.82 -34.58 301.52
```

.............................................................................................

**Function Signature:**
Write a function `f(inputs)` that processes the list of tuples where each tuple contains four real numbers.

```python
from typing import List, Tuple
def f(inputs: List[Tuple[float, float, float, float]]):
    '''
    inputs: a list of tuples, where each tuple contains four real
    ↪   numbers
    '''
```

---

Figure 2: An example of an open-ended problem description from our dataset.

the average diversity across the entire dataset:

$$\text{AvgDiv}_m = \frac{1}{N} \sum_{i=1}^{N} \text{Div}_m(P_i). \tag{1}$$

One key challenge is ensuring that our diversity metrics are invariant to the number of samples. When analyzing diversity among only well-formed and syntactically correct programs (see "Coherence" in Section 4), the number of samples can vary across instances. In such cases, a naïve strategy, such as calculating the proportion of semantically unique generations divided by the sample size, can lead to incorrect conclusions if sample size is not properly accounted for. Figure 3 illustrates this effect: using the naïve strategy (left), smaller samples yield significantly higher diversity scores than larger ones. To address this issue, we adopt a technique from Tevet & Berant (2021) to ensure invariance to sample size. Let $(p_1, p_2)$ be two input programs, and let $m_{\text{dist}} \in \mathbb{R}$ be a measure of distinctness or diversity. The diversity score for model $f$ on a problem description $x_i$ is then computed as the average over all unordered pairs in the multiset $P_i$ of LLM generations:

$$\text{Div}_m(P_i) = \frac{1}{\binom{|P_i|}{2}} \sum_{p_j, p_k \in P_i, j > k} m_{\text{dist}}(p_j, p_k). \tag{2}$$

Although this metric was initially motivated by other considerations, we find that it effectively addresses our sample size issue. In Appendix A.4, we prove that the metric does not depend on the sample size $n$ when $n$ is sufficientlylarge.

**Dataset and execution environment.** Several desiderata guide the creation of our dataset of problem descriptions. First, while the tasks should be open-ended, they must also be standardized enough to allow execution using consistent test cases and to enable comparison of outputs. Additionally, the tasks should be simple enough for both pre-trained and instruction-tuned models of varying sizes to generally solve.

We constructed our dataset by manually adapting competitive programming-style problems into abstracted programming tasks. In Figure 2, we show an example problem description from our dataset. For each problem description in our dataset, we provide an "Input Description" in natural language specifying the input format, an "Example Input" demonstrating potential inputs the function would handle, and a "Function Signature" providing a concrete specification of the function name and typing hints for the inputs. Importantly, we aggressively abstracted all program descriptions, removing any direct reference to the programming task in the description, standardizing the function name to $f(...)$, and using generic argument names.

We used competitive programming problems from CODENET (Puri et al., 2021) and accompanying test cases from ALPHACODE (Li et al., 2022) as a starting point for our dataset. Specifically,

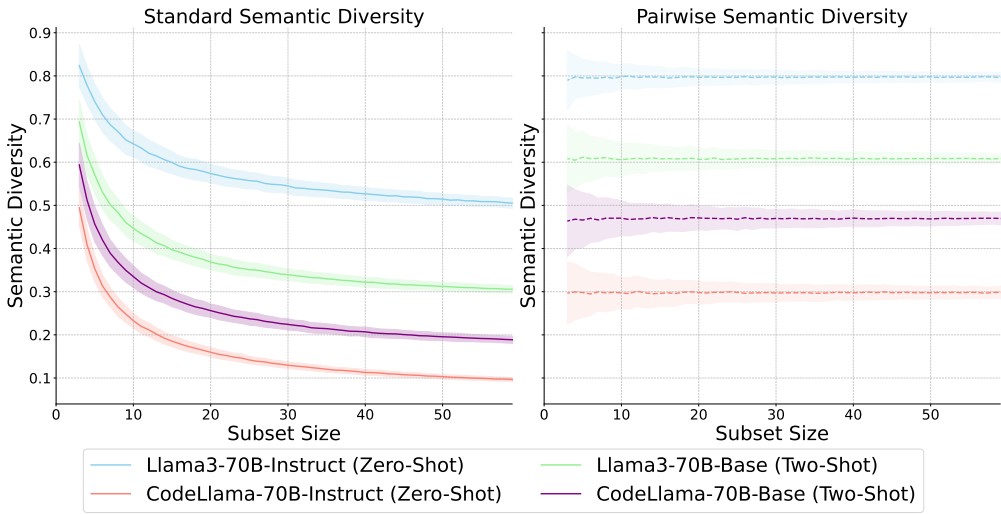

Figure 3: **Sample size confounds naïve diversity metrics:** For each sample size $S$ on the $x$-axis, we randomly sample $S$ programs and compute both a naïve diversity metric (left) and the pairwise diversity metric (right) and plot the mean and standard deviation across 500 random seeds.

we selected 21 competitive programming problems from CODENET and we abstracted problem descriptions into a canonicalized and open-ended format. Further details about the test set, its creation, and our execution environment are provided in Appendix A.1.

**Prompt selection.** The choice of prompt can significantly affect generation. Furthermore, LLMs that are not fine-tuned for instruction-following may struggle to generate *any* coherent programs without prompts that provide examples (Brown, 2020) or elicit chain-of-thought reasoning (Wei et al., 2022). To address this, we created three separate prompt templates: a zero-shot prompt, a two-shot prompt, and a two-shot prompt with chain-of-thought reasoning. This design allows us to probe generation behavior across a variety of settings. The few-shot examples included in the prompts were simple, manually written examples shared across all problems in the dataset. We provide the examples in Appendix A.6

## 4 DIVERSITY METRICS

Next, we describe the metrics used to measure diversity, focusing specifically on our choice of pairwise distance $m_{\text{dist}}(p_j, p_k)$.

**Semantic diversity.** We define two programs as semantically distinct if their outputs differ across a given set of test cases: $\mathbb{1}(O(p_j) \neq O(p_k))$, where $O(p)$ denotes the vector of outputs obtained by executing the test cases on program $p$. If the execution of $p$ results in an error for a particular test case, the test case output is recorded as the error. We allow two programs to be considered incorrect in different ways—for example, a syntactic error and a type error are treated as distinct. Note that if the generated programs fail to implement the function $f(...)$ entirely, this metric penalizes diversity, as such generations are all incorrect in the same way. In Appendix A.3, we analyze semantic diversity among only the subset of well-formed programs, using Equation (2) to reduce bias.

**Lexical diversity.** We use Expectation-Adjusted Distinct $n$-grams (**EAD**), an adaptation of the Distinct-N metric that removes bias towards shorter sequence length (Liu et al., 2022; Kirk et al., 2023). The Distinct-N metric (Li et al., 2016; Du & Black, 2019) computes the ratio between the number of *unique* $n$-grams divided by the *total* number of $n$-grams; in our case, we apply this to the combined text of the two generated programs. We tokenize programs using the Parso Tokenizer,[1] which allows tokenization of Python in the presence of syntax errors. We report EAD using $n$-

---

[1] https://github.com/davidhalter/parso

grams of length $n=4$. For our lexical and syntactic diversity metrics, we approximate Equation (2) by randomly sampling 300 pairs with replacement for efficiency.

**Syntactic diversity.** We adapt the Distinct-N metric to the Abstract Syntax Tree (**AST**) of each generated program. To further isolate the syntactic structure of a program (e.g., for-loop instead of recursion) from superficial choices (e.g., variable names), we canonicalize all identifiers and numeric constants in the AST, which we call the **Canonicalized AST**. We implement the Distinct measure on Canonicalized ASTs for two programs by calculating the ratio of the number of unique subtrees of height $H$ across both programs to the total number of subtrees of height $H$ in both programs, where $H=4$. In Appendix A.5, we provide a figure visually demonstrating a Canoncalized AST for a small Python expression and additional implementation details.

**Neural diversity.** We adapt existing methods of neural diversity metrics (Tevet & Berant, 2021) to our domain by using CODEBERTSCORE (Zhou et al., 2023) and ICE-SCORE, an LLM-based code-evaluation tool (Zhuo, 2024). For ICE-SCORE, we use `gpt-4o-2024-11-20` and the functional correctness setting. Since higher scores should indicate higher similarity, we use $1 - \text{Score}(p_j, p_k)$ for distinctness. While CODEBERTSCORE and ICE-SCORE were not intended for evaluating program diversity, we choose them since it either closely resemble models used in the NLP literature to evaluate semantic diversity (Tevet & Berant, 2021) or are state-of-the-art in neural code evaluation.

In our analysis, we report this number as the semantic diversity in the context of all samples taken: if a generation does not contain a program, we penalize the model as it does not produce semantically meaningful content.

**Coherence.** We additionally report a metric that measures the "quality" of generations. We use the term "coherence" to describe a well-formed generation with the following properties: (i) contains the definition of the function $f(...)$, (ii) has no syntax errors, (iii) is capable of being run on all test cases without runtime errors, and (iv) prints out any output (as requested by the prompt).

# 5 EXPERIMENTAL RESULTS

## 5.1 EXPERIMENTAL SETUP

**Models.** For open models, we use LLAMA-3, LLAMA3.1 (Dubey et al., 2024), CODE-LLAMA (Roziere et al., 2023), and QWEN-CODER-2.5 (Hui et al., 2024). A key benefit is that each one is instruction-tuned with different strategies—i.e., LLAMA-3 with a mix of PPO, DPO, Rejection Sampling, and SFT, LLAMA3.1 with a mix of DPO, Rejection Sampling, and SFT, QWEN-CODER-2.5 with DPO, and CODE-LLAMA with SFT only.[2] In addition to publicly available model checkpoints, we also created SFT-finetuned checkpoints for CODE-LLAMA, LLAMA-3, and LLAMA3.1 using MAGICODER's OSS-Instruct Instruction-Tuning Dataset (Wei et al., 2024). We fine-tune for two epochs with a learning rate of 3e-6, batch size of 32, a cosine learning-rate scheduler, and 300 warmup steps. For commercial models, we use `babbage-002`, `davinci-002`, `gpt-3.5-turbo-0125`, `gpt-3.5-turbo-instruct`, and `gpt-4o-mini` from OpenAI and SONNET 3 and HAIKU 3 from Anthropic.

**Sampling.** For each problem description $x_i$ in our dataset, we generate $K = 100$ programs, yielding 2,100 total programs sampled for each model $\times$ prompt pair experiment. For all Open Models, we used HuggingFace's `text-generation-inference`[3] on servers with 8 x NVIDIA RTX A6000 GPUs. For ANTHROPIC Models, we use the Amazon Bedrock API, and for OPENAI models, we use the OpenAI API. For all models, outside of Figure 1, we set the temperature to 1.0 without nucleus sampling.

**Statistical tests.** In Section 5.2, we evaluate the correlation between pairs of diversity metrics $m_1$ and $m_2$; to summarize these results, we report the Spearman and Kendall's Tau Rank Correlation coefficients of the values $\text{Div}_{m_1}(P_i)$ and $\text{Div}_{m_2}(P_i)$ aggregated across all sets of generations $P_i$ over all problem descriptions and all models.

---

[2]We tried to use models from ALPACAFARM (Dubois et al., 2024) given their clear documentation on instruction-tuning techniques used; however, they were incapable of generating coherent programs.

[3]https://github.com/huggingface/text-generation-inference

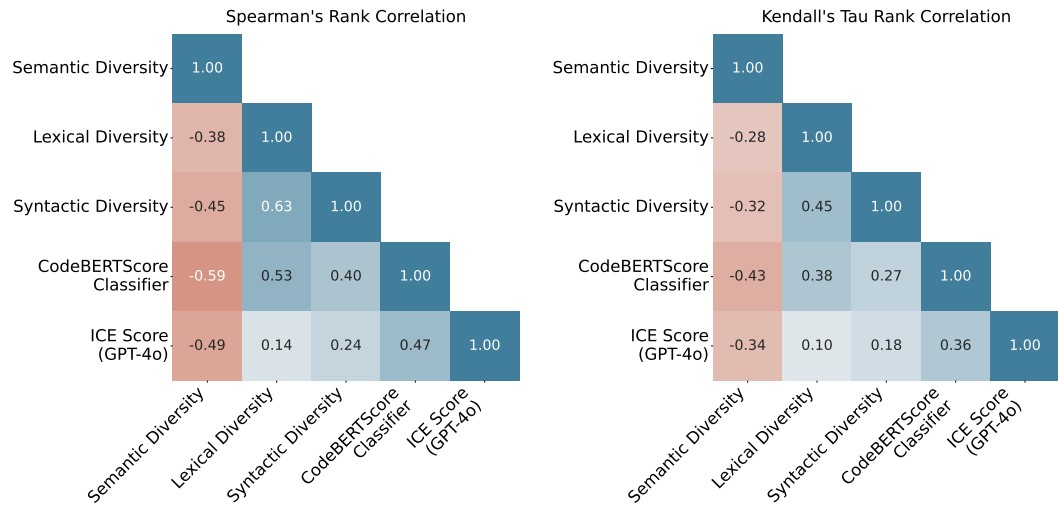

Figure 4: **Correlation between diversity metrics**: We analyzed the correlation of our diversity metrics across many experiments. We report the Spearman Rank Correlation and Kendall's Tau Rank Correlation Coefficients.

In subsequent experiments, we aim to determine if one kind of model increases or decreases diversity compared to another (e.g., instruction tuned or not). To summarize these results, we report the two-tailed Wilcoxon Signed-Rank Test (to measure significance) and Cohen's D (to measure effect size) for $\text{AvgDiv}_m$ over the entire dataset. A benefit of the non-parametric Wilcoxon statistical test is that we can make rigorous conclusions even if only limited samples are available We always pair models from the same family and vary whether the model is instruction-tuned, larger, or prompted with a different strategy *while fixing all other factors* unless otherwise noted. For example, when isolating model size, we would compare CODELLAMA7B-INSTRUCT with Zero-Shot prompting to CODELLAMA34B-INSTRUCT with Zero-Shot prompting, and so on. We use a paired test since models of one kind can be paired to natural counterparts of the other kind.

## 5.2 DIVERSITY METRICS CAN FAIL TO REFLECT EXECUTION SEMANTICS

```python
# CodeBertScore: 99.04
def f(A: int, B: int):
    print(A + B)

def f(A: int, B: int):
    print(A ^ B)
```

```python
# CodeBertScore: 98.78
def f(N: int, A: List[int]):
    print(min(A))

def f(N: int, A: List[int]):
    print(max(A))
```

Figure 5: Examples where CODEBERTSCORE fails to reflect execution semantics. Note we compare the program above to the program below. Both sets of programs are semantically different, but CODEBERTSCORE is very high. The similarity scores are in the top $1^{th}$ percentile for all programs and, despite being semantically distinct, is also in the top $1^{th}$ percentile of the wrong class (of all semantically equal programs).

**Existing diversity metrics fail to reflect execution semantics.** We report the Spearman and Kendall's Tau rank correlation coefficient matrices for diversity metrics in Figure 4. Our purpose is to motivate that diversity in program execution struggles to be captured by (and may even be negatively correlated with) lexical and neural diversity metrics. This is in line with prior work demonstrating that lexical and neural similarity metrics, at best, correlate relatively weakly with *functional correctness* (Hendrycks et al., 2021; Zhou et al., 2023; Zhuo, 2024), *however now for diversity*. We find that neural, lexical, and syntactic diversity fail to reflect semantic diversity based on executions accurately. All are negatively correlated. Additionally, lexical, syntactic, and neural diversity are

all inter-correlated to varying degrees. Given that neural metrics fail to reflect execution semantics, they did not offer insight into other forms of diversity for our task, so we omit them from subsequent experiments. In Section 5.3, we find a tradeoff in semantic diversity and lexical/syntactic diversity with instruction tuning. Given the large amount base and instruct models under consideration, these kinds of phenomena may contribute to the negative correlations.

**Program semantics may be more complicated to model than natural language.** Programming is a skill-intensive activity, and often, minor details can greatly impact program behavior. Whereas models can achieve over 90% accuracy on textual entailment tasks (He et al., 2021; Zhong et al., 2022), CODEBERTSCORE and ICE-SCORE report a moderate correlation with accuracy (Zhou et al., 2023; Zhuo, 2024). The complex nature of understanding programs combined with diverse and off-distribution generations may contribute to the relationships observed.

Anecdotally, for CODEBERTSCORE we found minor differences like a `min` instead of a `max`, and extra comments / unrelated generated code could trigger poor performance. Even though we may identify failure modes with our strict notion of semantics, this may be expected given CODE-BERTSCORE is not trained on execution semantics, and examples such as those in Figure 5 code are still related. Our takeaway is that we should not prima facie assume neural models can fully capture true semantic diversity: caution should used when evaluating diversity at scale, especially for programs.

## 5.3 EFFECT OF INSTRUCTION-TUNING ON DIVERSITY

| Model | N | Coherence | | Semantic | | Syntactic | | Lexical | |
|---|---|---|---|---|---|---|---|---|---|
| | | W (p) | ES (d) | W (p) | ES (d) | W (p) | ES (d) | W (p) | ES (d) |
| ALL OPEN | 36 | **<0.001** | 0.611 | **<0.001** | 0.788 | **0.001** | -0.743 | **0.001** | -0.572 |
| ALL OPEN (BEST COH.) | 6 | **0.028** | **1.073** | **0.028** | **1.609** | **0.028** | **-1.862** | **0.046** | **-1.414** |
| ALL OPEN (BEST SEM.) | 6 | **0.028** | **1.130** | **0.028** | **1.652** | 0.075 | **-1.467** | 0.116 | **-0.811** |
| CODELLAMA (**SFT**) | 9 | **0.039** | 0.579 | **0.020** | 0.454 | 0.164 | -0.749 | 0.203 | -0.278 |
| CL & ML W/SFT (**SFT**) | 18 | 0.130 | 0.245 | 0.119 | 0.359 | 0.899 | 0.159 | 1.000 | 0.248 |
| ML& QWEN (**PT**) | 18 | **0.001** | **0.958** | **0.002** | **10.240** | **<0.001** | **-16.182** | **<0.001** | **-14.786** |
| ML-3 (**PT**) | 6 | **0.031** | **1.979** | **0.031** | **2.154** | **0.031** | **-3.026** | **0.031** | **-1.834** |
| ML-3.1 & QWEN (**PT**) | 12 | **0.042** | **0.929** | **0.042** | **7.741** | **0.001** | **-17.719** | **<0.001** | **-16.288** |

Table 1: **Base vs. Instruct Comparisons.** Results from Wilcoxon's Signed-Rank Test p-values: **W (p)**, and Effect Size measured by Cohen's D: **ES (d)** with paired sample size as **N**. Bold p-values are below 0.05, and bold d-values have an absolute value greater than 0.8 (large effect size). BEST COH. and BEST SEM. are paired comparisons when choosing the best Model × Prompt pair for the given metric. **PT** indicates a preference-tuned model, **SFT** indicates supervised fine-tuning. Abbreviations CL and ML denote CODE-LLAMA and METALLAMA respectively.

**Less diversity in form, but more semantic diversity.** In Table 1, we summarize our results across all diversity metrics when comparing base models and their instruction-tuned counterparts. We find that across model sizes, prompts, and instruction-tuning methods, the general trend is that instruction-tuning; and furthermore, preference-tuning, increases coherence and semantic diversity and significantly decreases lexical and syntactic diversity. We also include two rows where we perform our paired tests for the best "Model × Prompt" group for both Coherence and Semantic diversity, discarding all other prompt pairs (e.g., we would compare LLAMA3-8B-BASE with Two-Shot prompting to LLAMA3-8B-INSTRUCT with Zero-Shot prompting if the prompts maximized coherence for the respective models). This adjusts for the potential that instruction-tuned models may behave differently than their base models for each prompt. After making this adjustment, the trend generally remains the same.

**The effect is stronger in preference-tuned models than in SFT models.** For the models fine-tuned with SFT, while the increase in semantic diversity and coherence is statistically significant for CODE-LLAMA, the effect sizes are moderate, the decreases in lexical and syntactic diversity are not significant. Furthermore, when considering the additional CODE-LLAMA, LLAMA-3, and LLAMA3.1 fine-tuned with SFT on MAGICODER's OSS-Instruct Instruction-Tuning Dataset, none

of the results are statistically significant. The results for these models could be due to the smaller SFT corpus or due to under-fitting. In contrast, this pattern is statistically significant for the preference-tuned LLAMA-3, LLAMA3.1, and QWEN-CODER-2.5 models and the effect size for semantic diversity ranges from very large to huge (Sawilowsky, 2009).

| Model | Base Model | | | | Instruction-Tuned | | | |
|---|---|---|---|---|---|---|---|---|
| | Coh. | Sem. | Syn. | Lex. | $\Delta$ Coh. | $\Delta$ Sem. | $\Delta$ Syn. | $\Delta$ Lex. |
| CODELLAMA-7B | 23.47 | 7.67 | 80.58 | 74.13 | 11.96 | 4.43 | -2.26 | -4.96 |
| CODELLAMA-34B | 26.56 | 10.24 | 80.00 | 74.06 | 14.51 | 5.14 | -0.28 | 0.36 |
| CODELLAMA-70B | 10.63 | 10.24 | 81.78 | 73.31 | 0.09 | 0.24 | -8.15 | -4.06 |
| META-LLAMA-3-8B | 25.18 | 9.05 | 84.26 | 71.97 | 47.29 | 22.81 | -15.13 | -6.94 |
| META-LLAMA-3-70B | 32.60 | 11.48 | 80.13 | 72.76 | 16.44 | 6.95 | -8.41 | -8.43 |
| META-LLAMA-3.1-8B | 8.31 | 8.19 | 84.30 | 68.48 | 11.51 | 11.43 | -16.29 | -3.73 |
| META-LLAMA-3.1-70B | 14.51 | 13.05 | 88.73 | 79.94 | 15.92 | 16.28 | -17.44 | -12.82 |

Table 2: **Comparison of Base Models vs Instruction-Tuned Models on Metrics**. We report the the metric and the difference for each model when choosing the "best prompt" according to the coherence score (positive values indicate that the metric increased).

In Table 2, we break down each model's results when taking the "best prompt" according to coherence. We see the relationship highlighted again: for the LLAMA-3 and LLAMA3.1 preference-tuned models, the preference-tuned models generally have much higher semantic diversity and more dramatic decreases in lexical and syntactic diversity.

## 5.4 EFFECT OF MODEL SIZE

| Comparison | Model | N | Coherence | | Semantic | | Syntactic | | Lexical | |
|---|---|---|---|---|---|---|---|---|---|---|
| | | | W (p) | ES (d) | W (p) | ES (d) | W (p) | ES (d) | W (p) | ES (d) |
| SMALL VS. LARGE | ALL OPEN | 18 | 0.108 | 0.151 | **0.001** | 0.515 | **0.038** | 0.353 | 0.108 | 0.287 |
| | BASE | 9 | 0.570 | 0.154 | 0.098 | 0.506 | 0.570 | 0.015 | 0.910 | 0.099 |
| | INSTRUCT | 9 | 0.203 | 0.180 | **0.012** | 0.688 | **0.004** | **0.817** | **0.039** | 0.522 |
| | COMMERCIAL | 18 | 0.579 | 0.237 | 0.212 | 0.425 | 0.284 | -0.496 | 0.495 | -0.282 |
| ZS VS. FS | ALL | 21 | 0.055 | 0.055 | 1.000 | 0.017 | **0.002** | -0.450 | **<0.001** | **-1.414** |
| | BASE | 7 | **0.016** | **2.575** | **0.016** | **7.020** | **0.047** | -0.794 | **0.016** | **-6.751** |
| | INSTRUCT | 7 | 1.000 | 0.087 | 0.297 | -0.525 | 0.109 | -0.619 | **0.047** | **-0.998** |
| | COMMERCIAL | 7 | 0.813 | 0.246 | 0.742 | -0.077 | 0.055 | -0.444 | **0.008** | **-1.201** |

Table 3: **Model Size and Zero- vs. Few-Shot Comparisons.** Results from Wilcoxon's Signed-Rank Test p-values: **W (p)**, and Effect Size measured by Cohen's D: **ES (d)** with paired sample size as **N**. Bold p-values are below 0.05, and bold d-values have an absolute value greater than 0.8 (large effect size). For Commercial models, because model size is unknown, we use cost-per-token as a proxy and match models within the same family to the best of our ability.

Next, we study how model size impacts coherence and diversity. In Table 3, we show the results when comparing small and large models of the same family.[4] For commercial models, model size is not transparent, so we use price per generated token as a proxy for size.

**Larger models increase semantic diversity without reducing the diversity of form.** We generally see higher semantic diversity in larger models without sacrificing lexical/syntactic diversity. This effect is more pronounced in instruction-tuned models, while results for base models are noisier and have smaller effect sizes. However, there is no statistically significant trend for commercial models, likely due to the variance in results across all pairs. If anything, there is weak evidence that more expensive commercial models may have less lexical and syntactic diversity than less expensive ones. We cannot draw stronger conclusions because these models are not well documented.

---

[4]We omit CODELLAMA-70B from being compared to the smaller models since the 70B base model and instruct-model were specialized with different pipelines than the smaller ones.

## 5.5 EFFECT OF ZERO-SHOT VS. FEW-SHOT

**Few-shot prompting increases semantic diversity in base models**. We find that few-shot prompts significantly increase both coherence and semantic diversity for base models. Similar to the effect of instruction-tuning on base models, we also see that few-shot prompting decreases lexical/syntactic diversity for base models.

**Instruction-tuned and commercial models lose lexical diversity**. Whereas there is no strong evidence that few-shot prompting impacts coherence or semantic diversity, there is strong evidence that it reduces lexical diversity for all models.

## 6 DISCUSSION AND CONCLUSION

**Contributions in methodology and empirical findings**. We have proposed a novel strategy for studying semantic diversity by focusing on code generation, where semantics can be automatically evaluated by executing programs on test cases. Using this methodology, we perform an extensive empirical analysis of LLM diversity. Our findings that all existing metrics (including neural metrics) correlate poorly with our semantic diversity metric motivate us not to assume these metrics reflect semantic content, especially for code. The result highlights the need for more rigorous methodologies for automatically measuring LLM diversity at scale.

**Our methods and results advance the discussion on LLMs and diversity**. We find that instruction tuning decreases lexical and syntactic diversity but increases semantic diversity; semantic diversity is not lost when accounting for coherence. Our empirical findings have *intellectual merit*. They add depth to the growing discussion on how instruction-tuning, model size, and prompting technique impact diversity. We speculate that if preference-tuning is associated with lower lexical/syntactic, it may impart a specific "voice" or style to the model. We hypothesize that a dynamic between the online and reward models used for rejection-sampling, PPO, and potentially DPO may induce a preference for certain lexical and syntactic constructs while preserving semantic diversity in specific domains like coding.

**Our empirical findings have practical consequences**. For creative endeavors and search-intensive tasks ranging from red-teaming LLMs (Perez et al., 2022), generating synthetic training data (Dubey et al., 2024), program testing Deng et al. (2024); Xia et al. (2024), and program optimization (Shypula et al., 2024), our insights may help others to decide which model configuration is optimal for both high-quality and diverse generations. For example, it may be more important to allocate resources to align smaller, cheaper, and faster models to the task with instruction-tuning than to allocate time and funds to sampling from a larger, slower, and more expensive model.

This result highlights the differentiated impact of instruction tuning on different kinds of diversity, suggesting that there may not be a "one-size-fits-all" strategy for preserving or improving diversity.

Finally, by the nature of our approach, we are restricted to evaluating code diversity. Code is an important domain in its own right; furthermore, for many tasks, code can be viewed as a formal representation of the natural language utterance (Liang, 2016), suggesting that our results may have implications for the diversity of natural language. Nevertheless, further study is needed to establish whether our findings translate to more traditional natural language tasks.

**Future Work**. While our work investigates the impact of instruction tuning, model size, and prompting on diversity, the nature of pre-training (e.g., the corpus and other training configurations) on diversity is an important direction for future work. We currently evaluate off-the-shelf neural models for their ability to reflect diversity in execution and note that other alternatives such as UNIXCODER (Guo et al., 2022) or CODEEXECUTOR (Liu et al., 2023) could have been chosen. We believe that efforts to make neural models more robust to capturing content diversity as well as more rigorous evaluations of neural models to measure LLM content diversity in natural language (for example, a human study following Tevet & Berant (2021) for LLM-generated content instead of human-generated content, is important. We also believe rigorously evaluating content diversity with more samples by modulating the degrees of open-mindedness in questions and across many domains and, from potentially sensitive domains to more harmless ones, will be important to understand patterns more broadly and the impact of efforts to reduce toxicity in LLMs.

## 7 REPRODUCIBILITY AND ETHICS STATEMENT

Upon acceptance of our work, we will release our dataset and the source code used to sample generations and evaluate diversity. We provide our dataset and driver scripts used for experiments in our Supplementary Material. We ensured that we saved all model generations used for our study, and we will publish the model generations related to all our experiments. We also constructed our program execution harness inside an isolated DOCKER container so that the reproduction of execution can be done safely for other researchers. We do not anticipate any risks in releasing our dataset and related code to the public.

No human subjects were involved in our work. While diversity in generation can be desirable, it is not always optimal, and practitioners should not overoptimize LLMs for diverse generations. For example, diverse generations may have adverse consequences for susceptible topics in natural language. In the code domain, diverse content must not entail the generation of malware or further enable bad actors to engage in cybercrime.

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

# A  APPENDIX

## A.1  DATASET CREATION AND ADDITIONAL DETAILS

We created our dataset through a multi-step process starting from the CODENET dataset and test-cases from ALPHACODE. The process involved problem standardization, language model assistance for scaling, and manual validation.

**Initial Processing.** We began by randomly selecting a single problem description from IBM CodeNet. We used this to as a seed to construct examples for a 1-shot prompt for a Language Model to assist us. Specifically we manually wrote one example of the following outputs that should be generated for each individual problem description from CODENET

1. A canonicalized problem description
2. A wrapper function that would take any generation for that function, parse inputs for the function, and instrument the generation with the entire suite of test cases
3. A property-based testing function for generating additional test cases when needed

**Dataset Expansion.** We then wrote prompt templates for `gpt-3.5-turbo-0125` that were designed to take our 1-shot prompt, and then prompt the LLM to repeat the same for the new example we select from CODENET. We randomly sampled 75 programs from CodeNet and attempted to generate the three components for each original problem description from CODENET. We then saved these into individual files, and also wrote them into an HTML document for manual review.

**Manual Review and Selection.** For the 75 problem ids that were processed, we then manually inspected the output components checking for the following criteria:

1. The language model correctly parsed inputs into our desired format
2. The problem added diversity to our dataset

We tracked the original CodeNet problem IDs and validation results in a spreadsheet. These were the 21 examples then used for our dataset.

**Manual Editing of Problem Descriptions.** After selecting our problems, we then manually went through each of the three components, and manually edited them to be correct: a large amount required revisions, as the LLM assistant made mistakes. We saved our manually edited components to be further processed.

**Test Case Integration.** For each of the individual problem descriptions, we then merged test cases from CODENET and ALPHACODE. We required at least 10 test cases per problem. For the three problems that lacked sufficient test cases, we used our property-based testing scripts to generate 100 additional cases, such that we would have sufficient coverage. We then saved the canonicalized problem descriptions, the function to extract and parse input test cases, and the input test cases into a dataset.

**Final Checks.** During experimental validation, we found and fixed one incorrect problem description and two faulty argument parsing functions. We then saved this corrected version as our final dataset.

**Additional Dataset Details.** In our experiments, we instruct LLMs to print their outputs to `stdout`; then, for each generation, we execute all test cases and capture the resulting outputs.

Because LLMs often generate natural language to accompany generated programs, extracting programs from the generations is a non-trivial task, especially for pre-trained models. We developed an extraction utility that extracts not only the target function $f(...)$, but also any helper functions and imports that may be relevant. To safely execute programs at scale, we perform all execution inside an isolated DOCKER container to prevent adverse consequences of blindly executing LLM outputs.

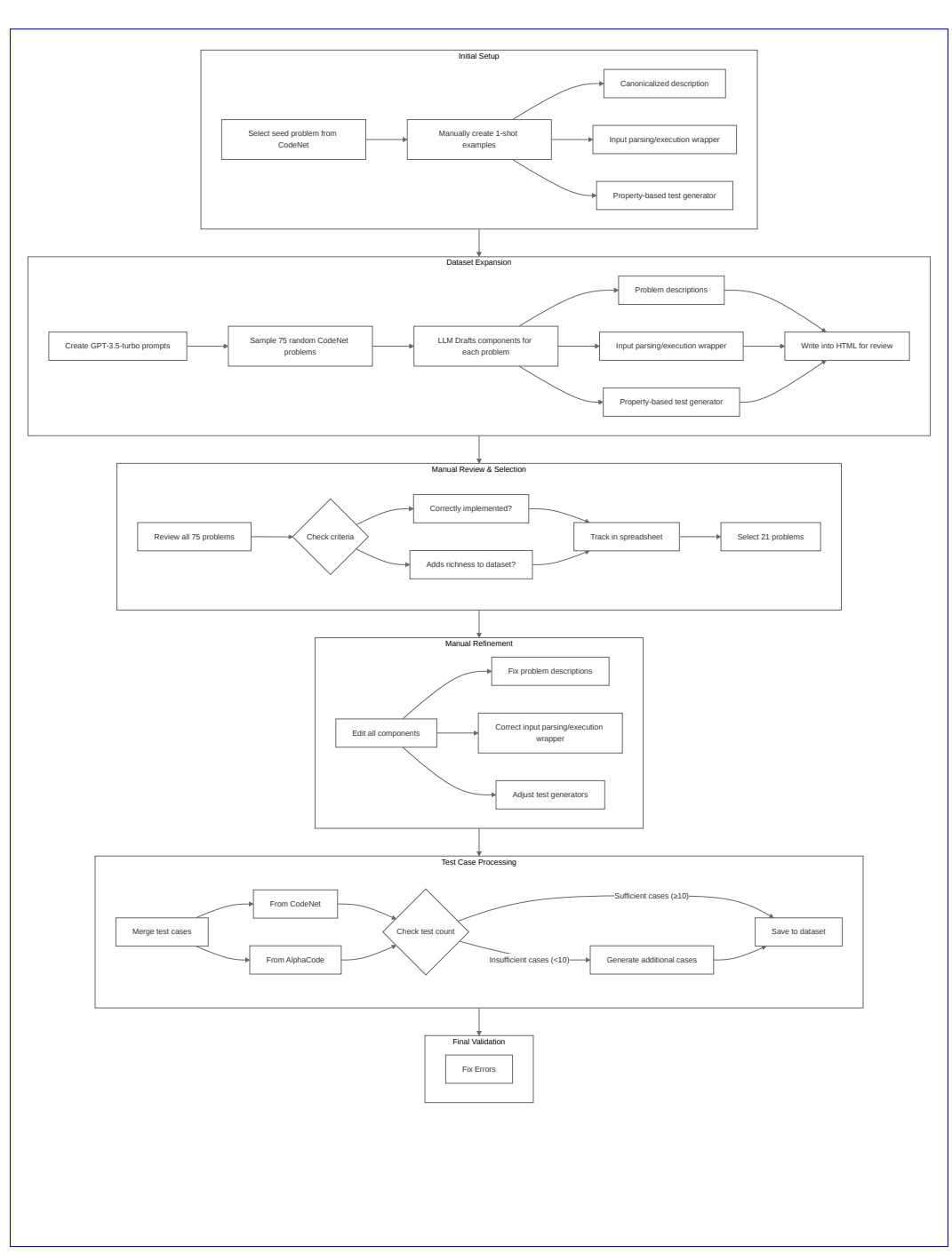

Figure 6: A flow-chart of the creation process showing the steps from initial problem selection through final validation. The process involved manual of 1-shot examples, `gpt-3.5-turbo-0125`, manual review and selection of diverse problems, refinement of problem descriptions and test cases, and final validation.

## A.2 MODEL CORRECTNESS BENCHMARKS

| Model | Size | Code Generation Benchmarks | |
| --- | --- | --- | --- |
| | | **HumanEval** | **MBPP** |
| CODE-LLAMA | 7B | 33.5 | 41.4 |
| | 34B | 48.8 | 55.0 |
| | 70B | 53.0 | 62.4 |
| CODE-LLAMA-INSTRUCT | 7B | 34.8 | 44.4 |
| | 34B | 41.5 | 57.0 |
| | 70B | 67.8 | 62.2 |
| CODE-LLAMA-PYTHON | 7B | 38.4 | 47.6 |
| | 34B | 53.7 | 56.2 |
| | 70B | 57.3 | 65.6 |
| LLAMA-3 | 8B | 37.2 | - |
| | 70B | 58.5 | - |
| LLAMA-3.1 | 8B | 8.5 | 47.6 |
| | 70B | - | 66.2 |
| LLAMA-3-INSTRUCT | 8B | 60.4 | 70.6 |
| | 70B | 81.7 | 82.5 |
| LLAMA-3.1-INSTRUCT | 8B | 72.6 | 72.8 |
| | 70B | 80.5 | 86.0 |
| Qwen2.5-Coder | 7B | 61.6 | 76.9 |
| | 14B | 64.0 | 81.0 |
| | 32B | 65.9 | 83.0 |
| Qwen2.5-Coder-Instruct | 7B | 88.4 | 83.5 |
| | 14B | 89.6 | 86.2 |
| | 32B | 92.7 | 90.2 |
| `code-davinci-002` | | 47.0 | 58.10 |
| `gpt-3.5-turbo-0125` | | 48.1 | - |
| `gpt-3.5-turbo-instruct` | | 68.0 | 82.0 |
| `gpt-4o-mini` | | 87.2 | - |
| Claude 3 Sonnet | | 73.0 | 79.4 |
| Claude 3 Haiku | | 75.9 | 80.4 |

Table 4: **Pass@1 scores on HumanEval and MBPP.** Results for the models as provided by Roziere et al. (2023) (CODE-LLAMA), Dubey et al. (2024) (LLAMA-3, LLAMA-3.1, Mistral, `gpt-3.5-turbo-instruct`), Hui et al. (2024) (Qwen2.5-Coder), Zheng et al. (2023) (code-davinci-002), OpenAI et al. (2023) (`gpt-3.5-turbo-0125`, `gpt-4o-mini`), and Anthropic (2024) (Claude 3).

## A.3 DIVERSITY WHEN CONTROLLING FOR COHERENCE

Generally, avoiding writing well-formed programs (i.e., without syntax and runtime errors) provides a good starting point for improving semantic diversity. In Table 5, we report results for the subset of generations without syntax or runtime errors.

**Semantic diversity does not increase.** When restricting to coherent programs, we find that instruction-tuning, increasing model size, and using few-shot prompting does not increase the semantic diversity. These results suggest that higher proportions of well-formed programs drive increases in diversity found above. Significantly as well, *we find no evidence* that instruction-tuning or larger models *reduce* semantic diversity.

| Comparison | Model | Size | Semantic | | Syntactic | | Lexical | |
|---|---|---|---|---|---|---|---|---|
| | | | W (p) | ES (d) | W (p) | ES (d) | W (p) | ES (d) |
| BASE VS. INSTRUCT | ALL OPEN | 21 | 0.708 | -0.051 | 0.083 | -0.527 | 0.083 | -0.548 |
| | ALL OPEN (BEST COH) | 6 | 0.075 | 0.079 | **0.046** | **-1.566** | **0.028** | **-1.608** |
| | ALL OPEN (BEST SEM.) | 6 | 0.345 | 0.030 | 0.116 | **-1.224** | **0.028** | **-1.276** |
| | CODELLAMA (**SFT**) | 9 | 0.570 | 0.040 | 0.426 | 0.474 | 0.570 | 0.386 |
| | LLAMA3 & LLAMA3.1 (**PT**) | 12 | 0.910 | -0.125 | **0.001** | **-1.886** | **0.005** | **-1.624** |
| | LLAMA3 (**PT**) | 6 | 0.688 | -0.268 | **0.031** | **-2.023** | 0.063 | **-2.143** |
| | LLAMA3.1 (**PT**) | 6 | 1.000 | -0.006 | 0.063 | **-1.606** | 0.063 | **-1.139** |
| SMALL VS. LARGE | ALL OPEN | 18 | 0.332 | 0.134 | **0.039** | 0.462 | 0.098 | 0.420 |
| | BASE | 9 | 0.401 | 0.238 | 0.176 | 0.176 | 0.461 | 0.130 |
| | INSTRUCT | 9 | 0.734 | 0.013 | 0.098 | **0.813** | 0.129 | 0.733 |
| | COMMERCIAL* | 18 | 0.890 | -0.109 | 0.454 | -0.470 | 0.330 | -0.381 |
| ZS VS. FS | ALL | 21 | **0.015** | -0.501 | 0.869 | 0.146 | 0.927 | 0.115 |
| | BASE | 7 | 0.688 | -0.308 | 0.156 | **1.336** | 0.094 | **1.353** |
| | INSTRUCT | 7 | 0.219 | -0.554 | 0.813 | -0.249 | 0.813 | -0.220 |
| | COMMERCIAL | 7 | **0.028** | -0.682 | 0.219 | -0.686 | 0.156 | -0.731 |

Table 5: **Restricting to Coherent Generations.** Results from Wilcoxon's Signed-Rank Test p-values: **W (p)**, and Effect Size measured by Cohen's D: **ES (d)** with paired sample size as **N**. Bold p-values are below 0.05, and bold d-values have an absolute value greater than 0.8 (large effect size).

**Preference-tuning reduces diversity of form.** We find that preference tuning reduces syntactic and lexical diversity, whereas syntactic and lexical diversity is somewhat greater in the SFT CODE-LLAMA models.

### A.4 ANALYSIS OF PAIRWISE DIVERSITY METRIC

For a given large language model (LLM) $f$, we assume that only a finite number $K$ of distinct semantic meanings can be generated by $f$. We first establish that the original semantic diversity metric converges to zero as the number of sampled responses tends to infinity. Specifically, the original semantic diversity metric is defined as

$$\frac{N}{n},$$

where $N$ is the number of distinct semantic clusters, and $n$ is the number of sampled responses. Since only a finite number of semantic meanings can be generated by $f$, the number of semantic clusters $N$ is bounded above, implying the existence of a constant $C_1 > 0$ such that $N \geq C_1$. Therefore, we have

$$\frac{N}{n} \geq \frac{C_1}{n} \quad \text{for all sufficiently large } n.$$

Now, observe that

$$\lim_{n \to \infty} \frac{C_1}{n} = 0,$$

so applying the squeeze theorem, we conclude that

$$\lim_{n \to \infty} \frac{N}{n} = 0.$$

Next, we show that the new metric defined in Equation (2), converges to a constant as $n \to \infty$. As before, we assume that there are $K$ distinct semantic meanings in total, and let $\pi_k$ denote the proportion of responses corresponding to the $k$-th semantic meaning. This implies that the number of times each semantic meaning is sampled is $\pi_k n$, where $\sum_{k=1}^{K} \pi_k = 1$. Thus, we have

$$\sum_{p_i, p_j \in P, i>j} m_{\text{dist}}(p_i, p_j) = \sum_{k \neq h} (\pi_k n) \cdot (\pi_h n) = \sum_{k \neq h} (\pi_k \pi_h) n^2,$$

where the summation is taken over distinct semantic meanings $k$ and $h$, and $m_{\text{dist}}(p_i, p_j)$ measures the semantic distance between generations. Moreover, the number of possible response pairs is

$$\binom{|P|}{2} = \frac{n(n-1)}{2}.$$

Thus, the new metric becomes

$$\frac{1}{\binom{|P|}{2}} \sum_{p_i, p_j \in P, i > j} m_{\text{dist}}(p_i, p_j) = \frac{2 \sum_{k \neq h} (\pi_k \pi_h) n^2}{n(n-1)} = 2 \sum_{k \neq h} (\pi_k \pi_h) + \frac{2 \sum_{k \neq h} (\pi_k \pi_h)}{n-1}.$$

Finally, we have

$$\lim_{n \to \infty} \left\{ 2 \sum_{k \neq h} (\pi_k \pi_h) + \frac{2 \sum_{k \neq h} (\pi_k \pi_h)}{n-1} \right\} = 2 \sum_{k \neq h} \pi_k \pi_h.$$

That is, as $n \to \infty$, the value of the new metric converges to the constant value:

$$2 \sum_{k \neq h} \pi_k \pi_h.$$

### A.5 ADDITIONAL INFORMATION THE SYNTACTIC DIVERSITY METRIC

We extract and process all Abstract Syntax Trees (ASTs) using Python's AST library with Python version 3.12.0, and report metrics for subtrees of height 4. Because syntactically incorrect programs do not parse, we can only calculate this metric over the subset of syntactically correct generations.

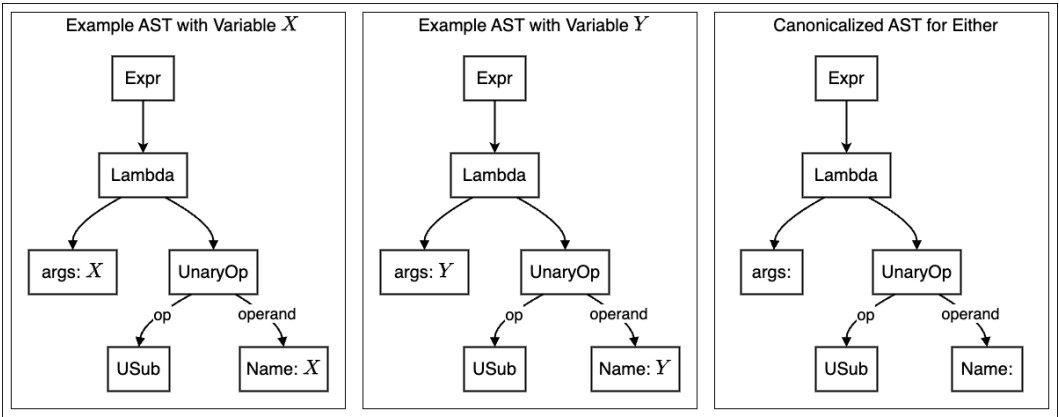

Figure 7: An Example of Canonicalizing an Abstract Syntax subtree used in the Distinct-CAST metric. The expression under consideration is for a simple lambda expression that negates a given variable. The first two ASTs are not equal because of the usage of the variables $X$ and $Y$, respectively, even though they are alpha-equivalent expressions. The AST on the far right canonicalizes identifier names such as arguments and variables so that both expressions would be equivalent.

### A.6 PROMPTS USED IN EXPERIMENTS

In Figure 8, Figure 9, and Figure 10, we show the prompting templates we use across all experiments.

```
{problem_description}

Now please implement the function f; do not return anything, the
↪  function f should print the result of the operation.
It should terminate within 30 seconds.
```

Figure 8: **Zero-Shot Prompt**: Our template for our zero-shot prompt, where the problem description would be input inside the curly braces.

```
### Input Description:
1. An integer \( N \) (1  \( N \)  10000), representing some
↪  quantity or size.
### Example Input:
```
1000
```
### Function Signature:
Write a function `f(N)` that takes in the input.
```python
def f(N: int):
    '''
    N: an integer
    '''
Now please implement the function f; do not return anything, the
↪  function f should print the result of the operation.
It should terminate within 30 seconds.
def f(N: int):
    print(n**2)
### Input Description:
1. A floating point number \( N \) (1  \( N \)  10000),
↪  representing some quantity or size.
### Example Input:
```
143.23
```
### Function Signature:
Write a function `f(N)` that takes in the input.
```python
def f(N: float):
    '''
    N: a float
    '''
Now please implement the function f; do not return anything, the
↪  function f should print the result of the operation.
It should terminate within 30 seconds.
def f(N: float):
    i = 0
    while N > 1:
        N = N / 2
        i += 1
    print(i)
{problem_description}
Now please implement the function f; do not return anything, the
↪  function f should print the result of the operation.
It should terminate within 30 seconds.
```

Figure 9: **Two-Shot Prompt**: Our template for our two-shot prompt, where the problem description would be input near the end inside the curly braces.

```
### Input Description:
1. An integer \( N \) (1  \( N \)  10000), representing some
↪  quantity or size.
### Example Input:
```
1000
```

### Function Signature:
Write a function `f(N)` that takes in the input.
```python
def f(N: int):
    '''
    N: an integer
    '''
Now please implement the function f; do not return anything, the
↪  function f should print the result of the operation.
It should terminate within 30 seconds. First describe the
↪  function you would write, then implement it.
The following function will print out the square of the input
↪  number. We will take the square using the ** operator in
↪  Python within the print statement.
def f(N: int):
    print(n**2)
### Input Description:
1. A floating point number \( N \) (1  \( N \)  10000),
↪  representing some quantity or size.
### Example Input:
```
143.23
```

### Function Signature:
Write a function `f(N)` that takes in the input.
```python
def f(N: float):
    '''
    N: a float
    '''
Now please implement the function f; do not return anything, the
↪  function f should print the result of the operation.
It should terminate within 30 seconds. First describe the
↪  function you would write, then implement it.
The following function will calculate the number of times the
↪  input number can be divided by 2 before it becomes less than
↪  1. We will increment a counter variable i each time we
↪  divide the number by 2 inside a while loop.
def f(N: float):
    i = 0
    while N > 1:
        N = N / 2
        i += 1
    print(i)
{problem_description}
Now please implement the function f; do not return anything, the
↪  function f should print the result of the operation.
It should terminate within 30 seconds. First describe the
↪  function you would write, then implement it.
```

Figure 10: **Two-Shot Chain-of-Thought Prompt**: Our template for our two-shot Chain-of-Thought prompt, where the problem description would be input near the end inside the curly braces.

