# OpenReview forum: "Does Instruction Tuning Reduce Diversity? A Case Study Using Code Generation"
_ICLR.cc/2025/Conference — Submitted to ICLR 2025_

### Official Review · Reviewer_ASEt · 2024-10-28

**Soundness:** 2
**Presentation:** 2
**Contribution:** 1
**Rating:** 1
**Confidence:** 4

**Summary:**

The authors conduct a case study on how instruction tuning affects the diversity of generated programs. Through the study, the authors find that instruction tuning reduces the lexical and syntactic diversity of generations but increases the semantic diversity of generations.

**Strengths:**

There are several strengths of the paper:
- The overall structure is clear and easy to follow.
- The authors conduct experiments on several Llama models.
- The authors evaluate the diversity from multiple dimensions.

**Weaknesses:**

The contributions and the setups are very limited, specifically:
- There is no evaluation of correctness metrics. Representative benchmarks like HumanEval [1] and BigCodeBench [2] should be included.
- Neural diversity like CodeBERTScore is not very appropriate here. Prior works like ICE-Score [3] show that CodeBERTScore is very weak compared to the evaluators based on LLMs as Judges. For example, the authors should consider using ICE-Score instead of CodeBERTScore.
- The conclusion that "Program semantics may be harder to model than natural language." is a bit wrong, as BERTScore itself is not robust [4-6]. There is no reason why CodeBERTScore should accurately reflect the program's correctness.
- The open models included in the evaluation are not diverse. The authors should use other Code LLMs, such as StarCoder2 [7] and CodeQwen [8].
- The training setups are not documented, and the results could be questionable.
- The experiments are Python-only. The authors should conduct evaluations on more programming languages.
- There is no related work on Code LLMs, code generation, and coding benchmarks.


[1] Chen, M., Tworek, J., Jun, H., Yuan, Q., Pinto, H. P. D. O., Kaplan, J., ... & Zaremba, W. (2021). Evaluating large language models trained on code. arXiv preprint arXiv:2107.03374.

[2] Zhuo, T. Y., Vu, M. C., Chim, J., Hu, H., Yu, W., Widyasari, R., ... & Von Werra, L. (2024). Bigcodebench: Benchmarking code generation with diverse function calls and complex instructions. arXiv preprint arXiv:2406.15877.


[3] Zhuo, T. Y. (2024, March). ICE-Score: Instructing Large Language Models to Evaluate Code. In Findings of the Association for Computational Linguistics: EACL 2024 (pp. 2232-2242).

[4] Hanna, M., & Bojar, O. (2021, November). A fine-grained analysis of BERTScore. In Proceedings of the Sixth Conference on Machine Translation (pp. 507-517).

[5] Sun, T., He, J., Qiu, X., & Huang, X. J. (2022, December). BERTScore is Unfair: On Social Bias in Language Model-Based Metrics for Text Generation. In Proceedings of the 2022 Conference on Empirical Methods in Natural Language Processing (pp. 3726-3739).

[6] Wang, J., Liang, Y., Meng, F., Sun, Z., Shi, H., Li, Z., ... & Zhou, J. (2023, December). Is ChatGPT a Good NLG Evaluator? A Preliminary Study. In Proceedings of the 4th New Frontiers in Summarization Workshop (pp. 1-11).

[7] Lozhkov, A., Li, R., Allal, L. B., Cassano, F., Lamy-Poirier, J., Tazi, N., ... & de Vries, H. (2024). Starcoder 2 and the stack v2: The next generation. arXiv preprint arXiv:2402.19173.

[8] Bai, J., Bai, S., Chu, Y., Cui, Z., Dang, K., Deng, X., ... & Zhu, T. (2023). Qwen technical report. arXiv preprint arXiv:2309.16609.

**Questions:**

- What's the column N in the tables?
- There is a lack of motivation to conduct investigations on code generation over text generation. The authors should provide sufficient explanations.

---

> ### Author Response · Authors · 2024-11-22
> **Response to Reviewer ASEt**
>
> We sincerely thank you for your thoughtful evaluation and for acknowledging the strengths of our paper in terms of structure, experimentation, and the multidimensional evaluation of diversity. We appreciate the opportunity to address some concerns and clarify any misunderstandings.
>
> ## 1. Undocumented Training Setups
> **Reviewer Comment:** *"The training setups are not documented, and the results could be questionable."*
>
> We apologize for any confusion. We believe there is a core misunderstanding.
> - **Clarification:** We utilized off-the-shelf models without any additional training. We will make sure this is clear in the paper to avoid misunderstanding.
>
> ## 2. Lack of Correctness Metrics Evaluation
>
> **Reviewer comment:** *”There is no evaluation of correctness metrics. Representative benchmarks like HumanEval [1] and BigCodeBench [2] should be included.”*
> - **Clarification:** We believe there may be a core misunderstanding: in our work we only evaluate off-the-shelf LLMs (e.g. LLama3.1), we do not do any specific fine-tuning on our own. Generally in the works we cited for these models, correctness metric evaluation has already been reported (e.g. HumanEval). Our focus was to evaluate diversity rather than correctness. In addition to this, we introduce the metric *coherence* to measure the quality and well-formedness of programs: we can think of this exactly as a correctness metric except allowing for open-endedness. We use coherence to determine if increased diversity comes at the expense of quality (coherence) and we report coherence in all our tables.
> - **Question:** Given this clarification, would you still like us to report correctness metrics like HumanEval for these models? We could retrieve these metrics from the respective publications and include them in an appendix sections. Otherwise, we believe that the coherence metric measures the important dimension of quality and well-formedness.
>
> ## 3. Limited Diversity of Models Evaluated
>
> We agree that including more models would enhance our study.
> - **Action:** We will include CodeQwen in our evaluations to increase the diversity of our analysis.
> - **Clarification:** We are limited in including StarCoder2 because it does not provide separate pre-trained and instruction-tuned models, which is important for our experimental design.
>
> ## 4. Appropriateness of CodeBERTScore
> - **Clarification:** We believe that CodeBERTScore is an appropriate and current metric for our study, yet we will augment our study with ICE-Score. We chose CodeBERTScore, it was the most analogous model to those used in recent diversity works BertScore and Sentence-BERT [1][2], making it a suitable starting point for assessing neural diversity in our study. Additionally, given its recent publication (published in December, 2023) and has been widely downloaded and utilized in the community (>400K downloads), we thought it reasonable to begin with.
>
> [1] Tevet, Guy, and Jonathan Berant. "Evaluating the Evaluation of Diversity in Natural Language Generation." Proceedings of the 16th Conference of the European Chapter of the Association for Computational Linguistics: Main Volume. 2021
>
> [2] Kirk, Robert, et al. "Understanding the Effects of RLHF on LLM Generalisation and Diversity." The Twelfth International Conference on Learning Representations.
>
> While ICE-Score offers some advantages, it has primarily been evaluated for assessing functional correctness or usefulness rather than diversity. Moreover, ICE-Score's reported correlations with correctness are moderate (e.g., correlations < 0.35 on Python HumanEval).
>
> We also note that there is limited literature on using ICE-Score or similar LLM-based evaluators specifically for measuring diversity. Our choice of CodeBERTScore aligns with standard practices in the diversity evaluation literature, which often employs BERT-like models due to their established use and validation.
>
> **Action:** We will also try to include ICE-Score in our analysis if feasible, to offer a comparative perspective; but we may not complete these experiments by the rebuttal deadline. In the revised paper, we will expand discussion of the limitations of CodeBERTScore and acknowledge the potential of alternative metrics. Additionally, we believe that exploring and validating new models for diversity measurement and assessing functional correctness is an important direction for future work, and we aim to motivate further research in this area.

---

> ### Author Response · Authors · 2024-11-22
> **Response to Reviewer ASEt (Part 2)**
>
> ## 5. Experiments Limited to Python
>
> -**Reviewer Comment:** *"The experiments are Python-only. The authors should conduct evaluations on more programming languages."*
>
> - **Action:** While resource constraints limit our ability to conduct extensive experiments in other languages, we will openly discuss this limitation in the paper and suggest it as an avenue for future work.
> - **Clarification:** To our knowledge it is often the case that diversity in Natural Language is only evaluated in English and also that many properties of program synthesis research is only evaluated on one language (usually and often Python). For example HumanEval was Python-only [3] and the APPS dataset is Python-only [4]. Additionally, our work makes empirical contributions from our findings and a methodological contribution in leveraging program execution to measure diversity. Taken together, we believe there is a significant contribution.
>
> [3] Chen, Mark, et al. "Evaluating large language models trained on code." arXiv preprint arXiv:2107.03374 (2021).
>
> [4] Hendrycks, Dan, et al. "Measuring Coding Challenge Competence With APPS." Thirty-fifth Conference on Neural Information Processing Systems Datasets and Benchmarks Track (Round 2).
>
> ## 6. Suggestion Regarding Paragraph in Section 4.2
>
> **Reviewer Comment:** *“The conclusion that "Program semantics may be harder to model than natural language." is a bit wrong, as BERTScore itself is not robust [4-6]. There is no reason why CodeBERTScore should accurately reflect the program's correctness.”*
>
> **Clarification:** Perhaps there is a misunderstanding in terms of our motivation of this paragraph; we do not intend to highlight this as a major contribution. We try to provide a rationale and discussion for the results that we see. As we discuss in the paragraph, we make the same statement that you make about CodeBertScore, that it correlates relatively poorly with functional correctness: *“CODEBERTSCORE reports a moderate correlation with accuracy”* (line 329-330).
>
> However, this fact may not be obvious to researchers with a background on diversity measurement in natural language. In natural language, it may be actually more reasonable to use neural networks to measure diversity than in code: neural measures of diversity have been reported to be competitive with human evaluations of diversity in natural language [3]. In this paragraph, we are only trying to explain why the nature between CodeBertScore and diversity is different for code than it is for natural language. While it may be obvious to some of us, it is intended to help make sense of why this is the case for researchers less familiar with work in the code space.
>
> [5] Stasaski, Katherine, and Marti A. Hearst. "Semantic Diversity in Dialogue with Natural Language Inference." Proceedings of the 2022 Conference of the North American Chapter of the Association for Computational Linguistics: Human Language Technologies. 2022.
>
> - **Action:** In the revised paper, we will discuss the limitations of current metrics like CodeBERTScore in accurately reflecting program semantics, and how this impacts the evaluation of diversity. We will also try to change language such that it is more clear this is not a contribution per-se, but rather a rationalization / explanation. If you like, we can remove this paragraph from the manuscript, however, we’re hesitant to do so if it will make the paper less appealing to a broader audience or if other reviewers object. If the reviewers agree that removing this paragraph is beneficial, we are happy to make this change.
>
> ## 7. Related Work Section
>
> Thank you for this suggestion.
>
> - **Action:** We will expand our related work section to include recent studies on code language models, code generation, and code benchmarks, situating our work within the broader research context.
> - **Clarification:** We originally prioritized the framing of our work as a paper within the research context of the diversity literature, and so we chose to prioritize this first.
>
> ## 8. Questions
>
> - **Column N in Tables:** The column 'N' represents the sample size available/used for statistical tests.
>
> - **Motivation for Code Generation over Text Generation:** We will enhance our motivation section to explain the significance of studying diversity in code generation, including practical applications and theoretical implications.

---

> > ### Comment · Reviewer_ASEt · 2024-11-22
> >
> > Thanks for the response.
> >
> > Regarding "The training setups are not documented", it should be corrected as "The inference setups". Sorry for the confusion.
> >
> > > Given this clarification, would you still like us to report correctness metrics like HumanEval for these models?
> >
> > I think reporting the functional correctness of these models will make the findings more concrete.

---

> > > ### Author Response · Authors · 2024-11-23
> > >
> > > Thanks for getting back! We will be sure to add more documentation regarding our inference setup. Regarding correctness, we can certainly report HumanEval scores for the models we evaluate, for example in a table in the Appendix so that readers can easily reference it; otherwise, with all the additional requests already made, we worry that we may not fit everything into the page limit. Do you think that is a reasonable approach?

---

> > > > ### Comment · Reviewer_ASEt · 2024-11-25
> > > >
> > > > Thank you!
> > > >
> > > > Please include the additional results in the Appendix.
> > > >
> > > > The primary concern with the current paper is that it focuses exclusively on analyzing generation diversity without making solid assertions about correctness. This could lead to conclusions that may be invalid or unsupported.

---

> > > > > ### Author Response · Authors · 2024-11-26
> > > > >
> > > > > Thank you for getting back to us. We will certainly include HumanEval scores in our Appendix for the underlying models. We are glad to know that this was your primary concern with the paper and that we will be able to remedy it.
> > > > >
> > > > > **Additional Clarification:** We would also like to bring up the use of *coherence* which we may consider as a correctness score (Section 3, lines 247-250).
> > > > >
> > > > > "*We additionally report a metric that measures the “quality” of generations. We use the term “coherence” to describe a well-formed generation with the following properties: (i) contains definition of the function f(...), (ii) has no syntax errors, (iii) is capable of being run on all test cases without runtime errors, and (iv) prints out any output (as requested by the prompt).*"
> > > > >
> > > > >
> > > > > In our opinion, *coherence* can be interpreted similarly to *accuracy.* In our experiments, we still enforce constraints: programs must be syntactically correct, they must correctly understand the prompt to generate a function and understand the input type, and they must process this input type without triggering a runtime error (e.g. divide by zero, indexing into a part of a list that is out of range), and they must also make use of the `print()` function to output the results to the console.
> > > > >
> > > > > We always intended that other accuracy metrics like HumanEval be used in addition to diversity and coherence to make judgements about LLMs (e.g. which LLM to use for a certain application). And we will be happy to include accuracy information in the paper, because we understand the importance it could have.
> > > > >
> > > > > The purpose of coherence was to understand how model size or preference tuning also impacts the well-formedness of generations. It is important to track this, especially to monitor if LLMs are incapable of respnding to ambiguous / open-ended tasks. For example, here is a quadrant that reflects how we very loosely and informally think about the coherence / diversity tradeoff.
> > > > >
> > > > > ```
> > > > > Coherence vs Diversity Matrix (informal and conceptual)
> > > > > ---------------------------
> > > > >                  Low Diversity   |   High Diversity
> > > > >
> > > > > High Coherence | Mode collapse   |   Ideal output
> > > > >                |                 |
> > > > >                | Well-formed but |   Diverse, generally
> > > > >                | repetitive      |   well-formed solutions
> > > > > ---------------|-----------------|------------------
> > > > > Low Coherence  | Poor output     |   Spurious output
> > > > >                |                 |
> > > > >                | Often erroneous |   Usually erroneous,
> > > > >                | not varied      |   can be varied
> > > > > ```

---

> > > > > ### Author Response · Authors · 2024-11-30
> > > > >
> > > > > Hello, we wanted to update you on the draft we have re-uploaded and to follow up on our previous discussion before the close of the discussion window. We’ve highlighted important changes to the paper in blue. We’d also like to mention that given the time crunch, (ICE-Score experiments, evaluating QWen-Coder, fine-tuning and evaluating models for reviewer kwb6’s comments) that there may be some typos that we’d certainly resolve in future drafts.
> > > > >
> > > > > Also we’d like to thank you for all your comments and feedback you’ve provided: we think our paper emerged stronger. We know that reviewing can be time consuming, and we’ve appreciated your time.
> > > > >
> > > > > ## 1.Reporting on Correctness Metrics
> > > > >
> > > > > You requested that we add metrics like HumanEval; in the appendix on Page 17, we’ve provided the metrics for HumanEval and MBPP for nearly all models. We have results for 29 models for HumanEval and 26 for MBPP. We are still waiting for one extra result for LLama-3.1 70B, because it seems Meta did not report the base model’s score, and the [BigCode evaluation harness](https://github.com/bigcode-project/bigcode-evaluation-harness) did not handle GPU memory efficiently enough on our hardware. We’re working on a custom evaluation harness for this one model.
> > > > >
> > > > > ## 2. Adding Qwen-Coder 2.5
> > > > >
> > > > > In our responses we mentioned we may not be able to add additional results for QWen-Coder 2.5. We were indeed able to evaluate the 7B Base, 7B Instruct, 34B Base, and 34B Instruct and we updated Table 1 with results using QWen Coder. We look forward to updating the other tables for the other experiments using the results from Qwen-Coder as time permits. We found the results for Qwen Coder aligned with the findings from LLama3.1, as both models use DPO in their fine-tuning stage [1]. We appreciate the suggestion as we fully agree that adding more models would help add more robustness and validity to the paper.
> > > > >
> > > > > [1] Hui, Binyuan, et al. "Qwen2. 5-coder technical report." arXiv preprint arXiv:2409.12186 (2024).
> > > > >
> > > > > ## 3. Adding ICE-Score Evaluation
> > > > >
> > > > > We were able to add our ICE-Score evaluation to our paper; for it, we used the functional-correctness setting and the most recent version of GPT-4o. Because of the high api cost of GPT-4o and the quadratic samples necessary for a brute-force diversity evaluation, we sub-sampled 122,500 pairs from all possible combinations to make the updated Figure 4 containing ICE-Score. We found that ICE-Score does not reflect execution semantics, and that correlates most with CodeBertScore. We believe that this can be intuitive, given that the authors of ICE-Score report a <0.35 example level correlation of Score with functional correctness on HumanEval [2]. Thank you for the specific suggestion to add ICE-Score as we can understand it removes the uncertainty if the phenomena is only due to CodeBERTScore being a smaller Bert-like model. Additionally, we think these results are meaningful to the community: to our knowledge we will be the first work to evaluate the efficacy of LLM-based diversity metrics, and we find they may experience similar shortcomings of smaller Bert-like models which have been predominantly used in the diversity literature.
> > > > >
> > > > > [2] Zhuo, Terry Yue. "ICE-Score: Instructing Large Language Models to Evaluate Code." arXiv preprint arXiv:2304.14317 (2023).
> > > > >
> > > > > ## 4. Documentation for Sampling
> > > > >
> > > > > As we explained, in our earlier draft there was no training; however, following your suggestion, we added documentation for our sampling setup on lines 311-316. Additionally, because we added fine-tuning experiments with reviewer kwb6’s suggestion, we added those details on lines 304-308.
> > > > >
> > > > > ## 5. Wording in Section 4.2
> > > > >
> > > > > We explained clarifications on the intent of this section: the purpose of this section is mainly to convince other skeptical readers who may believe that neural diversity metrics are completely sufficient; especially, as neural correctness and diversity has higher correlations in natural language than in code. We have updated the wording in Section 4.2 to make this more explicit, and we’ve cited more prior work to also make this more explicit. We are open to any suggestions in terms of how we frame things to improve the presentation of this section as well.

---

> > > > > ### Author Response · Authors · 2024-11-30
> > > > >
> > > > > ## 6. Related Work Section
> > > > >
> > > > > We updated the related work section to include more recent studies on LLMs for code, code benchmarks, and we believe that this has improved our paper. Thank you for the suggestions! Initially we wanted to highlight more findings and to strive to be concise, but we agree that the paper is much better with a richer background and related work section. The paper does probe some very specific empirical research questions, and the additional background will be useful to readers in the future.
> > > > >
> > > > > ## 7. Regarding the Scope of Python
> > > > >
> > > > > We believe that we have also provided a thorough response to the point you brought up about programming languages: we cited a few examples of Python used as the only language for datasets in a few works and also brought up that many works in NLP only do experiments on English as well. We would also like to add that we thought it was important to focus on Python, because it is commonly adopted in the community, it is easier to approach for other researchers, and it allowed us to control variables and find robust findings in one language. We believe that it would be interesting future work to see if these findings also generalize to other languages and domains, especially low-resource languages where there may not be as much examples.
> > > > >
> > > > > ## 8. Motivation for Empirical Questions in the Paper
> > > > >
> > > > > We’ve also added more richness to the background and related work sections to help explain the significance of studying diversity in code generation and in both practical applications of programming as well as the general question of diversity and its importance in applications like RLHF.
> > > > >
> > > > > You last mentioned that your primary concern was related to correctness. We tried to provide clarification that we indeed evaluate a metric that is closely related to correctness (Coherence: does the LLM meet the constraints provided by the prompt, while still allowing open-endedness). And following the suggestion, we’ve provided a large report of HumanEval and MBPP scores for the models used in the paper. We believe that we have addressed all the concerns you have brought up during this rebuttal period. With the remaining time, we would be interested to know if there is anything else we may further clarify.

---

> ### Comment · Reviewer_kwb6 · 2024-11-23
> **Reviewer kwb6's Comment**
>
> > I think reporting the functional correctness of these models will make the findings more concrete.
>
> I second this.

---

### Official Review · Reviewer_LiwB · 2024-10-30

**Soundness:** 3
**Presentation:** 3
**Contribution:** 2
**Rating:** 6
**Confidence:** 3

**Summary:**

The authors proposed a novel strategy for studying semantic diversity by focusing on code generation, illustrating some interesting properties of the output diversity for LLM open-ended generation.

**Strengths:**

1. The semantic diversity and other diversity measurement is crucial for openended tasks to evaluate the generation of models in terms of creativity, randomness, etc.
2. The metric devised by the authors makes for measuring model output diversities.
3. The most interesting part is the insights coming from the experiment results, offering fresh arguments for openended code generation diversity.

**Weaknesses:**

The major concerns are the meaningfulness of studying diversity and the generalizability of conclusions to general-domain openended generations.

1. Why is it meaningful to study diversity for LLM open-ended generation? From my perspective, it may contribute to more effective solution searching if we scale up the inference computation (like o1). But this is not discussed in this work, so whether only studying the diversity is  meaningful remains uncertain.
2. Code generation is a special case in open-ended generation: it usually has a "correct" output. As the semantic diversity is measured by considering the model output, then why it matters when it generates diverse outputs while only one of them is correct? Isn't it true that we only care about whether it covers the correct output?
3. Whether the conclusions obtained generalizable to the general-domain openended generation? I think it's hard to evaluate, and at the same time, the answer might be no.

**Questions:**

Can you elaborate more on the benefit of higher response diversity other than some applications like creative writing?

---

> ### Author Response · Authors · 2024-11-22
> **Response to Reviewer LiwB**
>
> ## 1. Meaningfulness of Studying Diversity in Code Generation
>
> **Reviewer Comment:**: *”Why is it meaningful to study diversity for LLM open-ended generation? From my perspective, it may contribute to more effective solution searching if we scale up the inference computation (like o1).“*
>
> We agree that the practical benefits of diversity need to be clearly articulated in the paper. We will improve on our presentation and writing. And you raise an excellent point about improving LLMs like O1 or scaling up inference-time compute - this was actually one of our initial motivations in measuring diversity.
> The importance of how we study diversity extends across several key areas:
>
> ### Immediate Practical Applications:
>
>
> - Enhancing best@K inference optimization (a form of scaling inference time compute as you noted)
>
>
> - Supporting/Enhancing model bootstrapping for models like O1 with better “exploration.” More diverse generations / reasoning trajectories are necessary for improving models with RL. Whereas, if models lack diversity, it is hard to bootstrap for improvement. Towards this, measuring semantic diversity can be used while tuning foundation LLMs to debug/monitor for instability or if mode collapse occurs. Additionally insights from our work can have practical implications for which model configurations and sampling strategies improve the pareto frontier exploration/exploitation trade-offs: if preference-tuned LLMs are indeed capable of high semantic diversity, we can feel confident sampling from them when trying to bootstrap for models like O1.
>
>
> - In programing: Generating comprehensive test suites that cover different edge cases and test for different program behavior
>
>
> - In programming: Handling ambiguous user specifications in real-world programming tasks. Here are some examples:
>
>   * For example if we ask an agent to try to draft a website N times, we want all N to be meaningfully diverse and high quality
>
>   * If we ask an agent to improve our code with N suggestions, we want all N suggestions to be diverse, high quality, and insightful
>
>   * If we ask an agent to perform a data science task with a certain dataset, there are numerous ways of pre-processing the data, fitting different models, and ensembling different models. Asking an agent to provide “insights” from data is even more open-ended; yet these are all programming tasks.
>
> In real-life applications of agents, humans will almost always provide vague instructions: agents must be capable of diverse generations. Our evaluation is supposed to measure the ability for LLMs to make “diverse generations with sufficient quality” which is not measured by normal programming tasks like HumanEval where there is only 1 “correct answer.” Unlike benchmarks like HumanEval, we make a contribution by creating a methodology to cheaply and systematically measure semantic diversity.
>
> ### Theoretical/Intellectual Understanding of LLMs:
>
> - It is important to know how instruction tuning affects different aspects of diversity. If we can measure diversity better, we can judge if LLMs are capable of diverse generation like humans. We believe that as ML researchers, this is a worthy end of research in its own right, without needing practical application.
>
> In our work we challenge existing understanding about the effects of preference tuning on diversity: compared to prior-work which argues preference-tuning reduces diversity, we find that instruction-tuning actually increases semantic diversity. And traditional diversity metrics breakdown, because when comparing instruction-tuned and preference-tuned models to pre-trained base models, not only are traditional diversity metrics poorly correlated… they’re negatively correlated.
>
> - **Action:** We will enhance the motivation section to discuss the importance of diversity in code generation, including applications in creative problem-solving, code synthesis, and improving model robustness.
>
> - **Examples:** We will provide concrete examples where multiple correct solutions are valuable, such as generating different implementations for ambiguous specifications or creating diverse unit tests.

---

> ### Author Response · Authors · 2024-11-22
> **Response to Reviewer LiwB (Part 2)**
>
> ## 2. Generalizability and Broader Impact
>
> **Reviewer Comment:** *”Whether the conclusions obtained generalizable to the general-domain openended generation? I think it's hard to evaluate, and at the same time, the answer might be no.”*
>
> While we're careful not to over-claim in our paper, we believe our findings have important implications beyond code generation. Here's why:
>
> - Our N-gram diversity results align with previous work in other domains (e.g. natural language summarization)
>
> - The discovery that preference tuning increases semantic diversity while reducing diversity in form (lexical/syntactic diversity) is plausibly a fundamental pattern in how instruction tuning affects model behavior. Anecdotally this resonates with our experience that preference-tuned models have a certain “voice” or “style” but are often capable of generating diverse content. More future work should be done to systematically evaluate these questions.
>
> - Our insights challenges existing assumptions and opens new research directions/problems for studying diversity in other domains
>
> Do you think it would be good to add reasons why our paper could have more generalizability beyond code to our discussion section? We would also want other reviewers to be on board as well.
>
> ## 3. Importance of Diversity When One Correct Output Suffices
>
> **Reviewer Comment:** *”Code generation is a special case in open-ended generation: it usually has a "correct" output. As the semantic diversity is measured by considering the model output, then why it matters when it generates diverse outputs while only one of them is correct? Isn't it true that we only care about whether it covers the correct output?”*
> - **Clarification:** Please correct us if we have not understood your question. Many real world programming tasks have multiple correct outputs as we mention in the previous response: we gave examples in data science, website design, and writing unit tests. However, these programming tasks generally have not been reflected in benchmark design. Diversity is crucial in helping address ambiguity in tasks, exploring different algorithms/techniques, optimizing for various constraints, and providing alternative solutions that may be more efficient, readable, or in better style.
> - **Correctness in our Paper:** In our paper, we introduce the term *coherence* to measure the quality of our generations. If a program is syntactically correct, if it encounters no type error or runtime error, and it prints output to the console, we consider the program coherent. We like to think of *coherence* similar to accuracy, but it allows for multiple semantically different answers that all are sufficient.
> - **Action:** We will include this discussion in the paper to highlight the significance of diversity even when “correctness” is achieved.
>
> ## 4. Reviewer's Questions
>
> - **Benefits of Higher Response Diversity Beyond Creative Writing:** As mentioned above, we will elaborate on practical applications of diversity in programming, such as improving inference-time compute, supporting ambiguous user requirements, and facilitating more comprehensive testing.

---

> ### Author Response · Authors · 2024-12-01
>
> Thanks again for reviewing our paper! We thank you for your recognition of our soundness, noting the significance of our methodological contribution of semantic diversity, and that the insights from the experimental results are interesting. We have uploaded an updated draft to OpenReview that also addresses the feedback you had and some clarifications we made. Key changes are highlighted in blue. While we've been engaged in extensive discussions with all reviewers and have conducted additional experiments requested by others (including evaluating LLM-based diversity metrics and adding QWen-Coder 2.5), we’d like to highlight the changes most relevant to your feedback:
> ## 1. Background and Related Work on the Importance of Studying Diversity
> Thank you for your suggestion to explain more of why it is meaningful to study diversity for LLM open-ended generation. We revamped our background and related work section (Section 2) to provide a much stronger background on the context in which we write our work, and we explain that there are a few crucial reasons to focus on diversity. Diversity is crucial for exploration, which is a necessary condition for improving models with RL or DPO, and the future of stronger LLMs. Additionally, as we discussed with you, there are numerous practical applications and real-world reasons why diversity matters: in the updated paper we mentioned extracting insights from data science, proposing website front-ends, scientific discovery, and software testing as a few examples of open-ended tasks that are important.
> ## 2. Broader Impact and Generalizability
>
> In terms of the broader impact, in the paper we also explain that diversity is important for improving LLMs with RLHF or DPO, and that our work challenges the prevailing narrative about preference tuning uniformly reducing diversity [1].
> As we explained in our comment to you, we personally believe that our results may generalize, but we are careful not to over-claim in our paper. We added to Section 6 a section on Future Work outlining different directions for future research that we believe would be interesting for exploration. Additionally, we’ve added experiments where we evaluate ICE-Score, an LLM-based approach for measuring program correctness [2], and also added QWen-Coder 2.5 as another preference-tuned model to lead to more robust results. These additional models followed patterns similar to those made in our original claims. We believe the new additions will help improve the robustness of our results and the validity of our findings.
>
> [1] Kirk, Robert, et al. "Understanding the Effects of RLHF on LLM Generalisation and Diversity." The Twelfth International Conference on Learning Representations.
>
> [2] Zhuo, Terry Yue. "ICE-Score: Instructing Large Language Models to Evaluate Code." Findings of the Association for Computational Linguistics: EACL 2024. 2024.
>
> ## 3. Importance of Diversity Instead of Improving Accuracy with Best@K Sampling
> We hope that our last comment resolved some questions and we also believe that our expanded background and related work section motivate that existing LLM program evaluation benchmarks do not evaluate diversity. We also hope that the additional discussion in the paper’s Background section about real-world applications, how diversity is important for exploration in RLHF, and prior work in the diversity literature also help explain why diversity and open-ended settings are important. Lastly, we would also like to add an additional point, that in our work, because of our use of the Coherence Metric and the Diversity Metrics, we can separate the effects of quality from diversity to analyze them (Section A.3 in the Appendix). Whereas with Best@K sampling, it may be difficult to disentangle the effects of higher quality models from models capable of diversity in open-ended or ambiguous scenarios.
>
>
> We also hope that in our response to you, we have clarified all questions you have had. If they are not clear, we would like to use the remaining time we have to provide additional clarity. We've worked diligently to address concerns from all reviewers, and the constructive exchanges have helped strengthen the paper significantly. Your feedback about the motivation for our paper and important applications was particularly important for our background section. We believe we've addressed the core concerns raised among all reviewers, and we are still waiting to hear back from everyone. We would appreciate your thoughts on whether these changes adequately address the specific concerns you had in your review.

---

> ### Author Response · Authors · 2024-12-02
>
> I hope this message finds you well. As we approach the end of the review period, I wanted to gently follow up regarding our revised submission. We greatly appreciated your thorough initial review and have made significant changes based on your valuable feedback, including:
>
> - Expanding our background and related work section to better justify studying diversity in LLM generation
> - Adding concrete examples of practical applications in programming, data science, and website design
> - Including additional experiments with ICE-Score and QWen-Coder 2.5 to strengthen our findings and improve generalizability
> - Clarifying the importance of diversity beyond improving accuracy with higher inference time compute
>
> Given today is the last day for reviewer responses, we would be very grateful if you could take a look at our revisions and provide any final thoughts. Your insights would be particularly valuable in assessing whether we've adequately addressed your initial concerns.
> Thank you again for your time and thoughtful consideration of our work

---

### Official Review · Reviewer_7Jxp · 2024-10-31

**Soundness:** 3
**Presentation:** 2
**Contribution:** 2
**Rating:** 5
**Confidence:** 3

**Summary:**

This work explores the impact of instruction tuning on the ability of Large Language Models (LLMs) to create diverse outputs. For their empirical analysis, they focused on code generation tasks, which allow for automatic verification of correctness via code execution. To measure code diversity, they introduced a holistic evaluation that encompasses semantic, lexical, syntactic, and neural diversity. The key contribution of this study is a methodology for evaluating LLMs' capacity to generate diverse code outputs. Additionally, they investigated the relationship between model size and diversity metrics.

**Strengths:**

1. The research focuses on a relevant matter, as there are few established evaluation frameworks in place to assess the capabilities of large language models (LLMs) in terms of diversity.

2. The study is not only theoretical but also includes several empirical demonstrations, providing tangible evidence.

**Weaknesses:**

1. The dataset construction could benefit from more detailed explanations.
2. The overall presentation of the study requires improvement to enhance clarity.
3. The study's findings regarding the research question, "Does instruction tuning reduce diversity?", are not conclusive.
4. The contribution appears insufficient for a comprehensive paper, but it could form the basis of a shorter piece, such as a research agenda exploring this topic further.

**Questions:**

### Comments and suggestions on Soundness
The authors utilized 21 handcrafted abstractions derived from CodeNet, along with corresponding test cases sourced from AlphacaCode, as the foundation for the dataset. Unfortunately, further specifics are inadequately detailed in the appendix, making it challenging to evaluate the integrity of the data collection procedure.

### Comments and suggestions about Presentation
1. The overall paper outline needs improvements. For example, as the study requires of multiple concepts and terminology to set the background (e.g., approaches to measure LLMs diversity, types of LLMs, instruction-tuning types, NLP techniques), I suggest including a stronger Background/Related Work section in the main paper. As a second example, I suggest creating a flow chart figure that clearly paints the big picture followed to create the dataset.
2. Grammar and Narrative. The paper needs general proofreading to improve quality. These are a few typos that I took note of: ‘the the’ (line 389), ‘We’ (line 296).

### Questions
1. Why are you presenting this work as a case study?

---

> ### Author Response · Authors · 2024-11-22
> **Response to Reviewer 7Jxp**
>
> All your points are really thought out, reasonable and very constructive, and we really appreciate the care you’ve taken in reviewing.
>
> ## 1. Need for Detailed Dataset Construction Explanation
>
> **Reviewer Comment:** *”The dataset construction could benefit from more detailed explanations.”*
>
> - **Action:** We will provide a detailed description of our dataset construction process in both the main paper and the appendix.
>
> - **Addition:** A flowchart or diagram will be included to visually represent the dataset creation and evaluation pipeline.
>
> ## 2. Improvements in Presentation and Clarity
>
> **Reviewer Comment:** *”The overall presentation of the study requires improvement to enhance clarity.”*
>
> Thanks for the constructive suggestions regarding the presentation, we do want the present the paper as well as possible.
>
> - **Action:** We will reorganize the paper to include a stronger Background and Related Work section, thoroughly introducing all key concepts and terminologies.
>
> - **Language Refinement:** We will perform a comprehensive proofreading to correct typos and improve narrative flow.
>
> ## 3. Regarding Conclusive Findings
>
> **Reviewer Comment:** *”The study's findings regarding the research question, "Does instruction tuning reduce diversity?", are not conclusive.”*
>
> - **Clarification:** Let us clarify our contribution relative to the title: Our work provides strong empirical evidence that challenges the dominant narrative about preference tuning and diversity. Previously the dominant narrative was that preference tuning reduced diversity [1]. In our experiments we find strong evidence via systematic measurement of semantic diversity that preference tuning increases semantic diversity while still reducing lexical and syntactic diversity. This is a non-obvious finding, because it requires defining a semantic diversity metric. Our methodology is also noteworthy, because it can be reproduced cheaply by other researchers; whereas, manual human evaluation would be extremely costly to reproduce.
>
> [1] Kirk, Robert, et al. "Understanding the Effects of RLHF on LLM Generalisation and Diversity." The Twelfth International Conference on Learning Representations.
>
> This finding isn't just interesting - it suggests that previous work using only lexical or neural metrics may have missed important aspects of how instruction tuning affects generation diversity. We hope that we can convince you of this perspective, and perhaps our exposition did not clearly articulate this.
>
>
> - **Action:** We will strengthen our discussion to clearly highlight how our results provide evidence that challenges existing narratives about instruction tuning reducing diversity.

---

> ### Author Response · Authors · 2024-11-22
> **Response to Reviewer 7Jxp (Part 2)**
>
> ## 4. Degree of Contribution
>
> **Reviewer Comment:** *”The contribution appears insufficient for a comprehensive paper, but it could form the basis of a shorter piece, such as a research agenda exploring this topic further.”*
>
> *We respectfully believe that our work makes a significant contribution.*
>
> - **Clarification:** As noted in the previous paragraph where we clarify our contributions, our paper challenges the dominant narrative on how preference-tuning impacts the diversity of generations. It is non-trivial, because we introduce a methodology to measure semantic diversity that is substantially different from lexical diversity or neural diversity metrics (Section 4.2 *Diversity Metrics Can Fail to Reflect Execution Semantics*) and furthermore is a methodological contribution to the community that can be cheaply re-evaluated in the future or expanded to be even more comprehensive.
>
> - **Action:** We will try to articulate some of these points better in an updated draft.
>
> ## 5. Question Regarding "Case Study" Presentation
>
> - **Clarification:** We are not committed that we have to use “case study” in the title, and we will reconsider the language. After internal debate on this exact question, we chose it, because we believe that a case study is enough to show that the story of how preference-tuning impacts diversity is more complex than we thought. The dominant narrative is that preference-tuning reduces diversity, and we find a strong pattern that lexical/syntactic diversity and semantic diversity can actually be negatively correlated when comparing pre-trained to instruction-tuned models. However, we are careful to not over-claim that this is the case over all domains, hence the choice of “case study” for now. Unofficially I think that it is likely that more domains will behave similarly, but we believe that it is best for future work to address this.
>
> - **Action:** We will put more thought into articulating the framing of our study, possibly removing the term "case study" to better reflect the scope and impact of our research or explain this point further.

---

> ### Author Response · Authors · 2024-12-01
>
> Thank you again for your thoughtful initial review, we particularly appreciate your recognition of our work's relevance and the value of providing empirical demonstrations in a space with limited evaluation frameworks. We have uploaded an updated draft to OpenReview that addresses your feedback with highlighted changes in blue. While we've been engaged in extensive discussions with all reviewers and have conducted additional experiments requested by others (including evaluating LLM-based diversity metrics and adding QWen-Coder 2.5), I'd like to highlight the key changes most relevant to your feedback:
>
> ## 1. Dataset Construction Documentation
> - We've added comprehensive documentation of our dataset construction process in the Appendix (pages 15-16)
> - This includes a detailed flowchart visualizing the complete pipeline
> - These additions provide transparency into our methodology and enable better reproducibility
> - We believe this documentation better demonstrates the careful consideration that went into our data collection process
>
> ## 2. Paper Structure and Presentation
>
> - We've substantially expanded the Background and Related Work sections to better contextualize our research
> - The improved structure better frames our key contributions, particularly our finding that preference tuning produces an interesting trade-off: increasing semantic diversity while reducing lexical/syntactic diversity
> - This finding challenges the prevailing narrative about preference tuning uniformly reducing diversity [Kirk et al., 2024]
>
> ## 3. Contribution and Findings
>
> - We've clarified how our work advances the field by:
>    - Providing a reproducible methodology for measuring semantic diversity
>    - Demonstrating that our semantic diversity metric measures properties of LLM generations that are not reflected in Neural or Lexical diversity metrics
>    - Showing that previous conclusions about diversity reduction may need to be reconsidered and studied further
>
> We've worked diligently to address concerns from all reviewers, and the constructive exchanges have helped strengthen the paper significantly. Your feedback about presentation and dataset documentation was particularly valuable in improving the paper's clarity. We believe we've addressed the core concerns raised in all reviews, and we are still waiting to hear back across the board, and would appreciate your thoughts on whether these changes adequately address your specific concerns.

---

> ### Author Response · Authors · 2024-12-02
>
> I hope this message finds you well. As we approach the end of the review period, I wanted to gently follow up regarding our revised submission. We greatly appreciated your thorough initial review and have made significant changes based on your valuable feedback, including:
>
> - Expanding the dataset construction documentation with a detailed explanation and a flowchart
> - Restructuring the paper with enhanced Background/Related Work sections
> - Clarifying our contributions
> - Addressing the presentation issues you identified
>
> Given today is the last day for reviewer responses, we would be very grateful if you could take a look at our revisions and provide any final thoughts. Your insights would be particularly valuable in assessing whether we've adequately addressed your initial concerns.
> Thank you again for your time and thoughtful consideration of our work.

---

### Official Review · Reviewer_kwb6 · 2024-11-04

**Soundness:** 3
**Presentation:** 2
**Contribution:** 2
**Rating:** 3
**Confidence:** 4

**Summary:**

This paper examines the impact of instruction tuning (via supervised finetuning (SFT) or preference-tuning (PT)) on output diversity, using code as a medium for semantic diversity through test case execution. Despite limitations in experimental scope (e.g., limited problem set, comparison using different baselines for SFT and PT), the study presents three notable findings: (1) optimal temperature values for achieving semantic diversity in code, (2) both SFT and PT increase semantic diversity, and (3) PT achieves greater semantic diversity than SFT.

**Strengths:**

- Figure 1’s temperature plot reveals potential improvement over CodeT's [1] default T=0.8, suggesting that T=0.9 or T=1.0 may yield higher diversity. This deserves further elaboration.

[1] Bei Chen, Fengji Zhang, Anh Nguyen, Daoguang Zan, Zeqi Lin, Jian-Guang Lou, & Weizhu Chen (2023). CodeT: Code Generation with Generated Tests. In The Eleventh International Conference on Learning Representations.

**Weaknesses:**

- The experimental results are limited to 21 competitive programming problems, raising concerns about generalizability.
- The comparison between CodeLLama (based on LLama2) and LLama3(.1) for SFT vs. PT may be problematic, as the use of distinct model versions could affect the observed diversity differences. A controlled experiment using LLama3(.1)-based SFT would improve reliability.
- Instruction-tuned LLM diversity could be influenced by the training set’s diversity. It would be beneficial to include a controlled comparison of LLMs without instruction tuning and LLMs with tuning that vary in training set or trajectory diversity.
- The finding that execution-based diversity differs from lexical/syntactic diversity is unsurprising. Prior work (e.g., APPS [2]) has documented similar discrepancies between lexical-based (BLEU) and execution-based metrics. Additionally, Section 4.2 reiterates results seen in counterfactual code augmentation research [3,4]. Further, the statement on lines 327-338 about the difficulty of semantic differentiation in code contrasts with the use of code to study semantic diversity.
- CodeBERTScore is outdated. More current options include UniXCoder [5] or CodeExecutor [6]. Another approach could leverage LLM-based evaluation, such as in CHASE-SQL [7], which employs a fine-tuned LLM as a code-pair verifier.

[2] Hendrycks, D., Basart, S., Kadavath, S., Mazeika, M., Arora, A., Guo, E., Burns, C., Puranik, S., He, H., Song, D., & Steinhardt, J. (2021). Measuring Coding Challenge Competence With APPS. In Proceedings of the Neural Information Processing Systems Track on Datasets and Benchmarks.

[3] Cito, J., Dillig, I., Murali, V., & Chandra, S. (2022). Counterfactual explanations for models of code. In Proceedings of the 44th International Conference on Software Engineering: Software Engineering in Practice (pp. 125–134). Association for Computing Machinery.

[4] Hojae Han, Minsoo Kim, Seung-won Hwang, Nan Duan, and Shuai Lu. 2023. Intervention-Based Alignment of Code Search with Execution Feedback. In Findings of the Association for Computational Linguistics: EMNLP 2023, pages 2241–2263, Singapore. Association for Computational Linguistics.

[5] Daya Guo, Shuai Lu, Nan Duan, Yanlin Wang, Ming Zhou, and Jian Yin. 2022. UniXcoder: Unified Cross-Modal Pre-training for Code Representation. In Proceedings of the 60th Annual Meeting of the Association for Computational Linguistics (Volume 1: Long Papers), pages 7212–7225, Dublin, Ireland. Association for Computational Linguistics.

[6] Chenxiao Liu, Shuai Lu, Weizhu Chen, Daxin Jiang, Alexey Svyatkovskiy, Shengyu Fu, Neel Sundaresan, and Nan Duan. 2023. Code Execution with Pre-trained Language Models. In Findings of the Association for Computational Linguistics: ACL 2023, pages 4984–4999, Toronto, Canada. Association for Computational Linguistics.

[7] Pourreza, Mohammadreza, et al. "CHASE-SQL: Multi-Path Reasoning and Preference Optimized Candidate Selection in Text-to-SQL." arXiv preprint arXiv:2410.01943 (2024).

**Questions:**

- Could you elaborate more regarding the best semantic diversity in Figure 1?
- Could the number of problems be increased to enhance generalizability?
- Could SFT and PT be compared on identical base models, for instance, by adding an SFT baseline trained on LLama3.1-base?
- Could a controlled comparison be added for LLMs with and without instruction tuning, varying training set or trajectory diversity?
- Could CodeExecutor be adopted for neural diversity evaluation?
- Could an LLM-based neural diversity metric, such as in CHASE-SQL, be added?

---

> ### Author Response · Authors · 2024-11-22
> **Response to Reviewer kwb6**
>
> Thank you for your time in reviewing our paper, we appreciate the time you put into it and your feedback. Hopefully we can make some clarifications, and we also have some quick questions (e.g. on the appropriate SFT dataset) we’d appreciate a reply to which may affect changes you requested.
>
> ## 1.Open-Ended Dataset Size
> We understand the concern regarding the generalizability due to the number of open-ended prompts in our dataset.
> - **Action:** In the camera ready version, we propose to increase the number of samples per problem from 100 to 200, resulting in a total of 4,200 programs. This enhancement will improve the robustness of experiments and reduce variance. Additionally, we will include examples in the supplementary material to illustrate the extensive variability within individual problems
> - **Justification:** Our problems are open-ended and designed to allow a vast space of valid solutions, each capable of exhibiting significant diversity.
>
> For example, **Problem 02280** (prompt provided in the supplementary materials) in our dataset prompts the language model to generate a function that processes a list of nodes. This prompt allows for a *very diverse* set of generations. For example, when we looked at some of LLama3.1 70B Instruct’s submissions here were some examples of the diverse generations it created:
>
> - Find all articulation points in a graph using Tarjan’s Algorithm
> - Find the number of connected nodes in a graph using Union-Find
> - Print the number of nodes in a graph with multiple valid non-cyclical paths in a specific and certain manner
> - Print out the count of unreachable and reachable nodes from node 0 in the list
> - Print out all nodes in the graph using DFS
> - Check if a tree is a valid binary tree
> - Check if the structure is a binary search tree (with some additional explicit constraints)
> - Remove nodes / relations in a graph with certain cyclical properties
> - Find the maximum depth in a tree-like structure
>
> Given the time it takes to sample from so many language models, we do not believe that we can complete this by the rebuttal deadline. However, we don't anticipate our results to change dramatically.
>
> Given that the prompts are so abstract and open-ended and having manually constructed this dataset from CodeNet, we believe the abstracted prompts are already quite comprehensive, and we understand that may sound very unintuitive. However, most competitive programming problems rely on relatively simple data structures like lists of integers. Despite this limited set of possible input types, competitive programming problems can ask for an extremely wide range of programming solutions. This is why we think taking higher samples will still achieve this goal.
>
> ## 2. Comparison Between Different Base Models
> We acknowledge that comparing CodeLlama with Llama3-based models may introduce confounding factors due to differences in base architectures, and we have tried to mitigate this effect on our conclusions and findings. Nevertheless, we commit to an SFT experiment with LLama3.1 8B for further robustness.
>
> - **Mitigation:** To address this, we have conducted paired statistical tests and observed consistent trends, particularly for the preference-tuned models across different model families. This indicates that the effects of preference tuning on diversity are robust despite differences in base architectures.
>
> - **Action:** We will also conduct additional experiments by performing SFT on a Llama3.1-base model and compare it to Pre-Trained to ensure a fair comparison.
>
> - **Feasibility:** While we may not complete these experiments before the rebuttal deadline, we commit to including them in the revised paper.
>
> - **Action Item:** We propose using the Magicoder OSS Instruct fine-tuning dataset [1] for this SFT experiment, we would like to confirm that you believe this is a sufficient approach. We’d like to know as early as possible to give us as good a chance to turn around results within the window.
>
> [1] Wei, Yuxiang, et al. "Magicoder: Empowering code generation with oss-instruct." Forty-first International Conference on Machine Learning. 2024.
>
> - **Clarification:** We will revise our draft to acknowledge that our claims regarding supervised fine-tuning (SFT) on their own are not as strong due to this confounding factor. However, we believe that our findings for preference tuning (e.g., PPO/DPO), model size, and prompting are robust due to the use of multiple model classes.
>
> ## 3. Influence of Training Set Diversity
>
> We agree that training set diversity can impact model diversity.
>
> - **Action:** We will clarify in the paper that while retraining large models to vary training set diversity is beyond our current resources, our methodology provides a framework for such studies.
> - **Future Work:** We will suggest that future research could leverage our benchmark to investigate the effects of training data diversity on model outputs.

---

> > ### Comment · Reviewer_kwb6 · 2024-11-22
> > **Reviewer kwb6's Response**
> >
> > Thank you for your response.
> >
> > > Action: In the camera ready version, we propose to increase the number of samples per problem from 100 to 200, resulting in a total of 4,200 programs. This enhancement will improve the robustness of experiments and reduce variance. Additionally, we will include examples in the supplementary material to illustrate the extensive variability within individual problems
> >
> > Wasn't it 21 problems, not 100? I'm referring to lines 201-202:
> > ```
> > ... We chose 21 competitive programming problems from CODENET and manually abstracted the problem descriptions. ...
> > ```
> > I think 100 refers to the number of generated test cases, as explained in line 654:
> > ```
> > ... we randomly generated 100 additional test cases by writing a property-based testing ...
> > ```
> >
> >
> > > Action Item: We propose using the Magicoder OSS Instruct fine-tuning dataset [1] for this SFT experiment, we would like to confirm that you believe this is a sufficient approach. We’d like to know as early as possible to give us as good a chance to turn around results within the window.
> >
> > I think using the Magicoder OSS Instruct fine-tuning dataset is a reasonable choice.

---

> > > ### Author Response · Authors · 2024-11-23
> > >
> > > *"I think using the Magicoder OSS Instruct fine-tuning dataset is a reasonable choice."*
> > >
> > > Great! We'll try our best to turn this around.
> > >
> > > *"Wasn't it 21 problems, not 100? I'm referring to lines 201-202:"*
> > >
> > > Our apologies, this key detail was accidentally omitted from our manuscript during drafting. In Section 2, we explain that for each problem in the dataset, we take $K$ samples (line 145); in *Section 4.1 we intended to explain that we take 100 samples per-problem* in and we sincerely apologize for this omission and the oversight that led to this.
> > >
> > > As a result, for each individual experiment (e.g. LLama3.1-Instruct 8B with Zero-Shot prompting), we have 100 samples / problem and 21 total problems, yielding 2,100 total generations. As a result, given 100 generations for each individual problem, we can tell if the LLM is capable of a very high degree of diverse outputs, or if it only repeats the same subset of semantically identical programs.
> > >
> > > By contrast, the authors in [1] try to measure diversity from Chat-GPT only asking it to "write a joke," re-wording / peturbing this question in 10 different ways. In their approach, one interpretation is that they are posing only one open-ended problem or request (and certainly no more than 10) to evaluate only one model: Chat-GPT. However, in our work we have 21 distinct prompts, one hundred generations for each, and we evaluate across many models and prompting scenarios. We don't have to rely on costly human evaluation, we can evaluate different dimensions of diversity (e.g. Lexical vs. Semantic), and we're able to probe a few separate empirical questions as a result.
> > >
> > > [1] Jentzsch, Sophie, and Kristian Kersting. "ChatGPT is fun, but it is not funny! Humor is still challenging Large Language Models." Proceedings of the 13th Workshop on Computational Approaches to Subjectivity, Sentiment, & Social Media Analysis. 2023.
> > >
> > > Here is an example of **Problem 02280** that we discussed in our rebuttal; from this we take 100 samples per model for each different prompting experiment (e.g. Zero-Shot, Few-Shot). This enables us to have many semantically diverse generations as we enumerated above ranging from finding the articulation points in a graph to validating if the structure is a binary tree. As we describe in our paper, we then apply either a zero-shot, few-shot, or few-shot with Chain-of-Thought prompt structure to this open-ended problem description template and then provide this as a full prompt to the LLM.
> > >
> > > ```python
> > > ### Input Description:
> > >
> > > An integer \( n \) (1 ≤ \( n \) ≤ 25), representing some quantity or size.
> > > A list of \( n \) tuples, where each tuple contains three integers \( id \), \( left \), and \( right \):
> > > - \( id \) (0 ≤ \( id \) < n), representing a unique identifier.
> > > - \( left \) (−1 ≤ \( left \) < n), representing a relationship or connection.
> > > - \( right \) (−1 ≤ \( right \) < n), representing another relationship or connection.
> > >
> > > ### Example Input:
> > >
> > >
> > > 9
> > > 0 1 4
> > > 1 2 3
> > > 2 -1 -1
> > > 3 -1 -1
> > > 4 5 8
> > > 5 6 7
> > > 6 -1 -1
> > > 7 -1 -1
> > > 8 -1 -1
> > >
> > >
> > > ### Function Signature:
> > > Write a function `f(n, nodes)` that takes in the input:
> > > def f(n: int, nodes: List[Tuple[int, int, int]]):
> > >     '''
> > >     n: an integer
> > >     nodes: a list of tuples, where each tuple contains three integers
> > > ```
> > >
> > > - **Action:** We will certainly update Section 4.1 with this detail and be ensure to articulate how our setup allows for a wide array of semantically distinct programs, for example, a wide array of graph algorithms ranging from finding articulation points, maximum depth, etc. We take full responsibility for this omission, that was an oversight that we never intended to happen.
> > >
> > > Does this help clarify the points we try to make about the nature of the dataset and about increasing the number of samples per problem?

---

> ### Author Response · Authors · 2024-11-22
> **Response to Reviewer kwb6 (Part 2)**
>
> ## 4. Use of CodeBERTScore
>
>
> **Reviewer Comment:** *“CodeBERTScore is outdated. More current options include UniXCoder [5] or CodeExecutor [6]. Another approach could leverage LLM-based evaluation, such as in CHASE-SQL [7]”*
>
>
> We appreciate your suggestions regarding evaluation metrics and understand the importance of using appropriate tools to assess neural diversity.
>
>
> - **Clarification:** We respectfully believe that CodeBERTScore is an appropriate and current metric for our study, yet we will try to add at least another related model-based approach. We chose CodeBERTScore, it was the most analogous model to those used in recent diversity works BertScore and Sentence-BERT [1][2], making it a suitable starting point for assessing neural diversity in our study. Additionally, given its recent publication (published in December, 2023) and has been widely downloaded and utilized in the community (>400K downloads), we thought it reasonable to begin with.
>
>
> [1] Tevet, Guy, and Jonathan Berant. "Evaluating the Evaluation of Diversity in Natural Language Generation." Proceedings of the 16th Conference of the European Chapter of the Association for Computational Linguistics: Main Volume. 2021
>
>
> [2] Kirk, Robert, et al. "Understanding the Effects of RLHF on LLM Generalisation and Diversity." The Twelfth International Conference on Learning Representations.
>
>
> - **LLM-Based Evaluators like CHASE-SQL:** We acknowledge the potential of LLM-based evaluators for code assessment. However, CHASE-SQL is specifically designed for evaluating SQL code and may not be directly applicable to Python code generation, which is the focus of our work. Adapting CHASE-SQL to Python would involve substantial modification and validation, which is outside the scope of this paper. Lastly, it seems that Chase-SQL is still under review and we were unable to find a reference implementation on Github. Instead, we will try to utilize ICE-Score [3], which is recent and LLM-based, as reviewer ASEt suggested.
> [3] Zhuo, Terry Yue. "ICE-Score: Instructing Large Language Models to Evaluate Code." Findings of the Association for Computational Linguistics: EACL 2024. 2024.
>
>
> - **Regarding UniXCoder and CodeExecutor:** While UniXCoder (2022) is a valuable model for code representation, we found that CodeBERTScore (2023) to be recent, rigorously evaluated on functional correctness (CodeBertScore Section 4), and bears resemblance to models used in the literature [1][2]. CodeExecutor, on the other hand, is specialized for code execution tasks and, to our knowledge, has not been evaluated in terms of its correlation with functional correctness or diversity metrics for code generation.
>
>
> - **Action:** While we recognize that exploring additional models could provide further insights, including every possible model may not be practical within the constraints of this work. Given the short time of response, we will add analysis using ICE-Score as reviewer ASEt suggested. For our camera ready we can commit to adding analysis for either UniXCode or CodeExecutor, but we cannot guarantee that we will add these by the rebuttal deadline. Additionally, we will expand our related work section to discuss alternative models like CHASE-SQL, UniXCode, and CodeExecutor.
>
>
> - **Future Work:** We consider the exploration of alternative models and more specialized LLM-based evaluators for Python code diversity as valuable directions for future research.

---

> > ### Comment · Reviewer_kwb6 · 2024-11-22
> > **Reviewer kwb6's Response (Part 2)**
> >
> > > However, CHASE-SQL is specifically designed for evaluating SQL code and may not be directly applicable to Python code generation, which is the focus of our work. Adapting CHASE-SQL to Python would involve substantial modification and validation, which is outside the scope of this paper.
> >
> > What I meant was to employ LLM-as-a-judge, and CHASE-SQL was just an example. I agree with Reviewer ASEt that ICE-Score could be a better option over CodeBERTScore.

---

> ### Author Response · Authors · 2024-11-22
> **Response to Reviewer kwb6 (Part 3)**
>
> ## 5. Findings on Execution-Based vs. Lexical Diversity
>
> **Reviewer Comment:** *“The finding that execution-based diversity differs from lexical/syntactic diversity is unsurprising.”*
>
> We appreciate your observation regarding the N-Gram-based metrics and execution-based metrics for functional accuracy. We believe there may be a misunderstanding around the purpose of this section of the paper.
>
> - **Clarification:** While previous studies have highlighted that execution-based accuracy and N-Gram metrics like BLEU do not correlate perfectly, we believe there may be a misunderstanding here. Our intention was not to present our correlations as a novel finding but rather to use it as a motivation for our methodology and validation that our implementation does what it is supposed to do. We are introducing a novel methodology to measure diversity, and it is important that it measures something that N-Gram metrics and Neural Metrics do not. By quantifying the significant differences between these metrics, we aim to convince readers—especially those who might be significantly more skeptical—that execution-based evaluation measures something substantially different in code generation compared to N-Gram/syntactic metrics. Moreover, there is a significant difference between reference-based accuracy metrics where there is only one correct answer and our methodological proposal for execution-based diversity which allows for a wide range of acceptable solutions: it is novel to use execution traces from programs as a method to measure diversity and it is not exactly straightforward to implement.
>
>
> - **Contribution:** Our results strengthen the case for utilizing execution-based metrics when evaluating diversity. While prior work has focused on functional correctness, our study validates that the disparity between execution-based and lexical measures extends to diversity assessments. This underscores the limitations of relying solely on lexical or syntactic metrics and highlights the necessity of execution-based evaluation to measure semantic diversity in generated code. While this fact may be obvious to some of us, leveraging this advantage for diversity measurement is non-trivial. Our methodology is novel, and as a result, we believe this is an important contribution to the body of work on measuring LLM diversity where no equivalent methodology has been proposed.
>
> ## 6. Elaboration on Temperature Settings (Figure 1)
> “Strengths: Figure 1’s temperature plot reveals potential improvement over CodeT's [1] default T=0.8, suggesting that T=0.9 or T=1.0 may yield higher diversity.”
>
> - **Clarification:** Perhaps there is a slight misunderstanding about the motivation behind including the figure. In Figure 1, we observed that increasing the temperature leads to higher semantic diversity up to a certain point before outputs begin to degenerate in quality. We did not aim to make "the optimal temperature" parameter scope of our work, but we believe these are interesting directions for future work. Instead, we included this figure as a proof-of-concept to validate that our implementation of semantic diversity is sound. It captures the idea that there is a "sweet spot" where the temperature maximizes diversity without compromising the coherence of the generated code. Our findings indicate that semantic diversity metrics effectively capture this phenomenon, whereas N-gram or syntactic measures do not.
>
> - **Reasoning:** We agree that the question of optimal temperature sampling parameters is interesting: this was a major motivation for our work, but we later found that answering questions about model size and instruction tuning were more urgent to the community and where our findings were most interesting. A short answer to your question about CodeT's default of 0.8: the optimal sampling temperature for open-ended diversity may vary from model to model. Furthermore, sampling for diverse outputs may require a higher temperature than sampling for “correct” outputs as the nature is different. As a result, we think that our dataset could be an interesting tool for researchers interested in choosing optimal sampling parameters for open-ended generation tasks.
>
> - **Future Work:** We think that more detailed future work should be done to analyze how additional sampling parameters affects diversity such as nucleus sampling and top-p sampling: for our draft our primary focus was on the effects of instruction tuning, model size, and prompting on diversity, where we believe our findings are particularly insightful and break new ground. If you think it would further strengthen our paper, we could investigate more questions if you think that would be valuable for the camera ready version if they are feasible.
>
> - **Action:** In response to your suggestion, we will expand our discussion of temperature settings in the revised paper.

---

> > ### Comment · Reviewer_kwb6 · 2024-11-23
> > **Reviewer kwb6's Response (Part 3)**
> >
> > > Contribution: Our results strengthen the case for utilizing execution-based metrics when evaluating diversity. While prior work has focused on functional correctness, our study validates that the disparity between execution-based and lexical measures extends to diversity assessments. This underscores the limitations of relying solely on lexical or syntactic metrics and highlights the necessity of execution-based evaluation to measure semantic diversity in generated code. While this fact may be obvious to some of us, leveraging this advantage for diversity measurement is non-trivial. Our methodology is novel, and as a result, we believe this is an important contribution to the body of work on measuring LLM diversity where no equivalent methodology has been proposed.
> >
> > I got your point. Still, please cite APPS in the revised version and note that the finding is consistent with them.

---

> > > ### Author Response · Authors · 2024-11-23
> > >
> > > Absolutely, we'll cite APPS and other works that had similar findings. In our opinion, we think that would also strengthen the discussion in that section! I can completely understand the motivation behind this recommendation: given how the paper was drafted I can understand how it can be interpreted as missing or overlooking the findings of executable benchmarks like APPS that N-gram metrics can very poorly approximate functional correctness. That was truly never our intention, and I think maybe we took that fact for granted amongst ourselves given our familiarity with executable benchmarks. But I completely agree this is important to articulate, and is very constructive, because it helps frame the paper in the proper context.

---

> ### Author Response · Authors · 2024-11-22
> **Response to Reviewer kwb6 Individual Questions (Part 4)**
>
> **7. Questions**
> - **Best Semantic Diversity in Figure 1:** We have clarified what our takeaways for this were and we will update our draft to also expand the discussion around this point. We also believe that our dataset will enable important future work on these questions.
> - **Increasing Number of Problems:** As mentioned, we believe increasing samples per problem enhances the robustness of our experiments more effectively than adding more problems.
> - **Comparing SFT and PT on Identical Base Models:** As addressed above, we will include experiments comparing SFT and PT using the same LLama3.1. Additionally, we have explained that this only affects conclusions for SFT, and that we have already tried to mitigate for this effect by using more than one model family for PT as well as direct and paired comparisons for model size and prompting experiments.
> - **Controlled Comparison with Varying Training Set Diversity:** We will acknowledge this as an important area for future work and discuss how our benchmark could facilitate such studies.
> - **Adopting CodeExecutor for Neural Diversity Evaluation and Adding LLM-Based Neural Diversity Metrics:** We do not believe that Chase-SQL may be directly utilized for diversity evaluation without some custom modification and adaptation. We’ve mentioned that we used CodeBertScore as it was published in 2023 and bears high resemblance to models that have been evaluated for diversity. We will try to the best of our ability to evaluate another neural metric, ICE-Score as reviewer ASEt suggested by the rebuttal deadline, and at least one more for the camera-ready version.

---

> ### Comment · Reviewer_kwb6 · 2024-11-23
> **Reviewer kwb6's Response**
>
> As I reread lines 030-032:
> ```
> Many real-world tasks are open-ended to some degree with many possible answers—e.g.,
> writing convincing essays, suggesting cooking recipes, writing unit tests, etc. Evaluating diversity
> can also provide insights into the nature of language models, especially their creative capabilities.
> ```
> I think 'many possible answers' implies that there exist some constraints or conditions for expected outputs (as random word sequences cannot be considered valid answers).
> The main problem in this paper is that the vagueness in problem descriptions. This makes it hard to scale up because problems need to be manually annotated, and makes it difficult to evaluate functional correctness (as the problem allows different functionalities), leaving the evaluation metrics reliant on either human efforts or model-based alternatives which could be erroneous.
>
> One way to tackle the original question—evaluating the model's diversity—is to narrow down the constraints that the expected outputs would have. That is, if we fix the expected functionality, then we can ensure the validity of answers while measuring lexical and syntactic diversity through n-grams and AST-based similarity metrics. This direction is scalable, as we can directly employ existing code generation benchmarks (e.g., BigCodeBench [1] of 1K problems), and we can also ensure functional correctness.
>
> Compared to this possible direction, why should we use the proposed approach, which is not scalable (only 21 problems) and does not ensure the validity of outputs relative to the problem description?
>
> [1] Anonymous (2024). BigCodeBench: Benchmarking Code Generation with Diverse Function Calls and Complex Instructions. In Submitted to The Thirteenth International Conference on Learning Representations.

---

> > ### Author Response · Authors · 2024-11-25
> >
> > Thank you for letting us know about this new point that you brought up. We want to reply to you soon, but we wanted to give you a heads-up that we will be slightly delayed, because we are prioritizing the requested experiments and revising the paper draft right now, especially given the extension for the discussion period.

---

> > > ### Comment · Reviewer_kwb6 · 2024-11-25
> > > **Reviewer kwb6's Response**
> > >
> > > Personally, I would like to recommend prioritizing this fundamental question.
> > > > Compared to this possible direction, why should we use the proposed approach, which is not scalable (only 21 problems) and does not ensure the validity of outputs relative to the problem description?

---

> > > > ### Author Response · Authors · 2024-11-30
> > > >
> > > > We wanted to follow up and update you that we were able to update our draft of the paper to address a lot of the asks you had in your original rebuttal. We’ve highlighted important changes in the draft in blue; there may be slight typos in blue in case we missed things in a crunch. We would appreciate you bring up important comments you have before the window closes in case there are anything you would like to add that you may not have had the chance to bring up yet.
> > > >
> > > > More than anything we’d like to thank you for all the time and effort you invested in reviewing: reviewing can be a thankless process. But please know that we really appreciate all the time and effort you put into this and going above and beyond what I think is often the case for reviewers for ML conferences, not just for us but the community at whole. The back-and-forth has provided a lot of important points of improvement from a scientific perspective and it also helped us improve how we articulate important details about our paper .
> > > >
> > > > ## 1. We have added ICE-Score to our Evaluation of Neural Models of Diversity
> > > >
> > > > We were able to add our ICE-Score evaluation to our paper; for it, we used the functional-correctness setting and the most recent version of GPT-4o. Because of the high api cost of GPT-4o and the quadratic samples necessary for a brute-force diversity evaluation, we sub-sampled 122,500 pairs from all possible combinations to make the updated Figure 4 containing ICE-Score. We found that ICE-Score does not reflect execution semantics, and that correlates most with CodeBertScore. We believe that this can be intuitive, given that the authors of ICE-Score report a <0.35 example level correlation of Score with functional correctness on HumanEval [1]. We are glad for the suggestions to add additional neural models, as your suggestions here will also help the community understand if LLM-based evaluators of correctness and code diversity are significantly more robust or not than smaller models like CodeBERTScore: to our knowledge, no work has evaluated or utilized these very large LLMs for diversity metrics.
> > > >
> > > > [1] Zhuo, Terry Yue. "ICE-Score: Instructing Large Language Models to Evaluate Code." arXiv preprint arXiv:2304.14317 (2023).
> > > >
> > > > ## 2. We Have Added Comparisons Between Different Base Models
> > > >
> > > > We mentioned that we may not be able to complete the SFT experiments with LLama3.1 by the rebuttal deadline. We were in a crunch between attending to extended back and forth, but we were able to evaluate three SFT models: a CodeLlama, Llama3, and Llama3.1, and we’ve added all three of these to section 5.3 between lines 402 to 409. We’re happy you brought up this one suggestion, because we think it was a great idea and feasible. As we originally mentioned, any results from this would not impact our methodological contribution as well as what we consider our main results: finding that preference-tuning (i.e. DPO or PPO) increases semantic diversity, and reduces lexical/syntactic diversity. For SFT, we found that the added models weakened the statistical significance of the results, which were generally quite weak to begin with. We believe that the models we fine-tuned are inferior to Meta’s version of CodeLlama, and that these results should be taken with a grain of salt, due to the size of OSS-Instruct Dataset being smaller and less-comprehensive than the CodeLlama SFT datasets. Furthermore, we have not fully completed analysis of these results. While we cannot add the draft we are happy to commit to updating all tables and sections from any findings from these SFT models. We appreciate having both the SFT models we have as well as Meta’s version of CodeLlama, because we believe having Meta’s CodeLlama’s version can also remove a confounder from an insufficiently large or comprehensive SFT dataset.
> > > >
> > > > ## 3. Re-Wording and Clarification of Execution-Based vs. Lexical/Syntactic Diversity
> > > >
> > > > Following your suggestion, we tried to make our motivation for section 5.2 clear that our purpose is *motivation,* and also a demonstration in the domain of *diversity.* As we explained, our main reason is to convince a broader audience, especially those who are skeptical that execution is hard for neural models to reason about, that execution-based diversity measures something that neural models cannot capture. And we agree with your assessment that stronger citation of findings such as those in the APPs paper just strengthens this section. So we thank you for the constructive feedback!
> > > >
> > > > ## 4. Added Explanation of Figure 1: Temperature Sweep
> > > >
> > > > Following your suggestion, and in addition to our discussion, we added additional explanation of the motivation and significance of the temperature sweep into our introduction.

---

> > > > ### Author Response · Authors · 2024-11-30
> > > >
> > > > ## 5. Pre-Training Corpus as Future Work
> > > >
> > > > Following your suggestion, we added that we believe that changing the pre-training corpus is an important direction for future work. We also realized that given our SFT experiments, we could potentially compare CodeLLama and MetaLLama (as the pre-training corpus is different). We didn’t have time to explore the differences in results from our OSS-Instruct SFT experiments between CodeLLama and MetaLlama, but we are also interested in looking into this. And we also strongly agree that there may be many interesting research questions about how the pre-training corpus affects downstream diversity; however, this may suit research groups with a higher amount of resources in the future.
> > > >
> > > > ## 6. Previous Points of Discussion
> > > >
> > > > We have provided detailed responses regarding our choice of open-ended problems over "correct/incorrect" questions, as well as evidence for how our methodology places meaningful constraints on generation while allowing for future modifications to add additional constraints in the future. We are happy to discuss these points further if helpful.

---

> ### Author Response · Authors · 2024-11-26
>
> My response will be imperfect, because we're in a time crunch between the paper revision and other things happening this week. But I appreciate the thinking you're doing about the paper, they're very good questions. We hope that we could convince you that we have thought about similar questions and yet we strongly believe the work has substantial merit.
>
> ## 1. Choosing Open-Ended Questions Instead of "Directed" (i.e. either Correct or Incorrect) Questions
>
> *"That is, if we fix the expected functionality, then we can ensure the validity of answers while measuring lexical and syntactic diversity through n-grams and AST-based similarity metrics. This direction is scalable, as we can directly employ existing code generation benchmarks (e.g., BigCodeBench [1] of 1K problems), and we can also ensure functional correctness."*
>
> Actually, early on, we initially intended to analyze how lexical and syntactic diversity would change when looking at directed experiments where there is only 1 correct answer. We have working code to do this, and we are happy to provide these in our supplementary .zip if you like; we plan to open source them on acceptance.
>
> However, we didn't think that the questions we could ask with such a setup were going to make any strong methodological contributions nor noteworthy empirical findings. Methodologically, evaluating lexical and syntactic diversity is a weak contribution at best. Prima-facie, increasing/decreasing lexical or syntactic may not have a great degree of practical importance. And some preliminary work on natural language summarization (where the semantic content generally will be the same) has already probed the effects of preference-tuning on lexical diversity [1].
>
> We strongly believe *the most interesting contributions and research questions were related to open-ended generation*. This is because code can measure semantic diversity at scale. The questions about automatically differentiating semantic vs. diversity in form (lexical and syntactic diversity) is a methodological contribution and fills a missing need in the community. And the empirical questions we probe, the evidence we find that preference-tuning increases semantic diversity and decreases diversity in form (lexical and syntactic diversity), these challenge the dominant narrative on how preference-tuning (e.g. RLHF) impacts diversity [1]. As a result, we strongly believe the methodological and empirical findings are both noteworthy.
>
> [1] Kirk, Robert, et al. "Understanding the Effects of RLHF on LLM Generalisation and Diversity." The Twelfth International Conference on Learning Representations.
>
> ## 2. Potential Lack of Constraints
>
> *"I think 'many possible answers' implies that there exist some constraints or conditions for expected outputs (as random word sequences cannot be considered valid answers)...."*
>
> I appreciate you bringing up these points, and we've thought hard about similar points while working on this, and I can understand that without clarification I'd also have the similar questions.
>
> I would respectfully disagree, and strongly argue that our semantic diversity places constraints on programs. We think Figure 1 in the paper does a great job of showing how semantic diversity as we measure it drops when we increase Temperature to be very high. *As the generations become more like random strings the semantic diversity rapidly decreases*. This is because nearly all generations become syntactically incorrect, and therefore are incapable of meaningful semantic diversity. Our implementation captures this phenomenon. However, we see that as temperature becomes high, the lexical diversity becomes very high. *On the "temperature test" our semantic diversity is much more robust to the effect of random / spurious strings*
>
> In general, if all the generations are syntactically incorrect or the LLM consistently misunderstands the prompt and is not capable handling types in the right way, the semantic diversity will suffer. The syntactic constraints, type constraints, and the requirement to print results to the console *are constraints on outputs.* Generally, most of our models achieved Coherence (Section 3 for description) scores below 50%, implying that often LLMs are incapable of following the instructions in the prompt. We think future work can and should explore adding even more constraints on outputs, (e.g. requiring only 1 integer printed out between a certain range), but we did not feel it was necessary nor impacted the validity of the empirical questions we ask in our paper. For example, a major narrative in our paper is how preference-tuning impacts diversity. We believe we find enough evidence as it is to challenge the dominant narrative that preference-tuning reduces diversity: instead, we find it can reduce diversity in lexical form while increasing semantic diversity. Setting a low temperature value (e.g. 0.2) is much more likely to constrain semantic diversity than opting for preference-tuning.

---

### Author Response · Authors · 2024-12-04

As the review period ends, we would like to thank all the reviewers for their thoughtful feedback and time.

During our discussion period, multiple reviewers recognized the significant contributions of our work:

**Reviewer 7Jxp noted both the critical need and empirical rigor of our work:**
- *"The research focuses on a relevant matter, as there are few established evaluation frameworks in place to assess the capabilities of large language models (LLMs) in terms of diversity"*
-  *"The study is not only theoretical but also includes several empirical demonstrations, providing tangible evidence."*

**Reviewer LiwB emphasized both the methodological and empirical value:**
- *"The semantic diversity and other diversity measurement is crucial for openended tasks to evaluate the generation of models in terms of creativity, randomness, etc."*
-  *"The most interesting part is the insights coming from the experiment results, offering fresh arguments for openended code generation diversity."*

## *We have significantly strengthened the technical depth and rigor of our work through multiple substantial additions:*

### 1. Enhanced Evaluation Framework:

- We’ve added ICE-Score evaluations using GPT-4o as suggested by Reviewers kwb6 and ASEt. This adds more robustness to and further reinforces our findings on the nature of semantic diversity measurement vs. neural and lexical diversity metrics

### 2. Expanded Model Coverage:
- Incorporated QWen-Coder 2.5 (7B Base, 7B Instruct, 34B Base, 34B Instruct) providing additional validation of our key findings
- New SFT Experiments: Conducted new fine-tuning experiments on Llama3, Llama3.1, and CodeLlama-7b-Python using Magicoder-OSS-Instruct-75K
 - Comprehensive Correctness Metrics: Added extensive HumanEval and MBPP scores for models evaluated

### 3. Rigorous Documentation:

- Added 2-pages of detailed dataset construction documentation with flowcharts and thorough process explanations

These additions strengthen our key finding that challenges the dominant narrative about preference tuning (i.e. RLHF or DPO) reducing diversity:

- The QWen-Coder results align with our findings from LLama3.1, providing critical validation across model families
ICE-Score evaluations demonstrate that even advanced LLM-based metrics struggle to capture semantic diversity, reinforcing the value of our execution-based methodology

- Our new SFT experiments provide controlled comparisons and add further robustness by controlling for base-model comparisons

## *To our knowledge, we have systematically addressed major reviewer concerns that were brought up:*

- Shared concern (kwb6, ASEt) about neural metrics: Added ICE-Score evaluation using GPT-4o
- Reviewer kwb6's request for controlled SFT experiments: Completed with three new model variants
- Reviewer ASEt's concern about correctness metrics and model coverage: Added comprehensive HumanEval/MBPP results and QWen-Coder-2.5 results
- Reviewer 7Jxp's request for documentation: Added two pages of detailed methodology including flowcharts
- Reviewer LiwB's questions about motivation: Expanded background section with concrete applications and demonstrations about importance

Given the additional experiments requested, we intend to increase sample size per problem to 200 and naturally to ensure our langauge is polished. In our work, we challenge the dominant narrative that methods like RLHF or DPO universally reduce diversity. Our expanded experiments and analysis provide evidence that traditional diversity metrics may lead to incomplete conclusions. Like in prior work, we find that PPO/DPO reduces lexical diversity; however, we find that it may actually increase semantic diversity (diversity of meaningful content, and not just superficial form). These findings, supported by our novel execution-based methodology and comprehensive evaluations across multiple model families, represent an important contribution to our understanding of how instruction tuning affects model behavior.

---

### Meta-Review · Area_Chair_9br5 · 2024-12-19

**Metareview:**

The paper investigates how instruction tuning affects diversity in code generation outputs from large language models (LLMs). The main findings show that preference tuning can increase semantic diversity while reducing lexical/syntactic diversity, challenging existing beliefs about instruction tuning's impact on generation diversity. While the methodology of using code execution for measuring semantic diversity is novel, there are several important limitations: (1) experiments are restricted to only Python language and 21 problems, raising concerns about generalizability, (2) lack of comprehensive model coverage in initial submission, and (3) insufficient explanation of dataset construction process. These limitations, combined with presentation issues and questions about broader impact beyond code generation, suggest the work would benefit from substantial improvements before publication.

**Additional Comments On Reviewer Discussion:**

During rebuttal, authors added HumanEval/MBPP scores, QWen-Coder experiments, and ICE-Score evaluations as requested by reviewers. They also improved dataset documentation and background sections. However, key concerns about limited problem set, Python-only experiments, and generalizability to other domains remain insufficiently addressed despite author efforts to explain and justify their approach.

---

### Decision · Program_Chairs · 2025-01-22

Reject